# Position: Why is plausibility surprisingly problematic as an XAI criterion?

## Abstract

Explainable artificial intelligence (XAI) is motivated by the problem of making AI predictions understandable, transparent, and responsible, as AI becomes increasingly impactful in society and high-stakes domains. The evaluation and optimization criteria of XAI are gatekeepers for XAI algorithms to achieve their expected goals and should withstand rigorous inspection. To improve the scientific rigor of XAI, we conduct a critical examination of a common XAI criterion: plausibility. Plausibility assesses how convincing the AI explanation is to humans, and is usually quantified by metrics of feature localization or feature correlation. Our examination shows that plausibility is invalid to measure explainability, and human explanations are not the ground truth for XAI, because doing so ignores the necessary assumptions underpinning an explanation. Our examination further reveals the consequences of using plausibility as an XAI criterion, including increasing misleading explanations that manipulate users, deteriorating users' trust in the AI system, undermining human autonomy, being unable to achieve complementary human-AI task performance, and abandoning other possible approaches of enhancing understandability. Due to the invalidity of measurements and the unethical issues, this position paper argues that the community should stop using plausibility as a criterion for the evaluation and optimization of XAI algorithms. We also delineate new research approaches to improve XAI in trustworthiness, understandability, and utility to users, including complementary human-AI task performance.

## 1 Introduction

Data-driven predictive technologies, now primarily called artificial intelligence (AI) [77], have become impactful in high-stakes domains such as healthcare, finance, and criminal justice. This underscores the importance of the research field of interpretable or explainable AI (XAI) to provide reasons for AI predictions in human-understandable ways [22]. The key purposes of XAI are to provide users with informed decision support to understand the boundaries and error patterns of AI capabilities, empower users to question and challenge AI predictions to hold algorithms accountable [11], and improve the performance of the human-AI team by enabling users to identify potentially uncertain or flawed AI predictions to leverage the strengths of both [33, 40]. Achieving these purposes can make the AI system more understandable, transparent, and trustworthy. Explainability is also one of the five AI ethics principles that enables other four principles of beneficence, non-maleficence, autonomy, and justice through intelligibility and accountability [26].

However, deploying an XAI algorithm does not automatically guarantee explainability or its benefits unless the XAI passes rigorous validation. Otherwise, the XAI is suspected of ethics washing [85, 3, 46, 35]. Evaluation methods are then vital to safeguard the scientific and responsible development of XAI, and should withstand critical examinations. To improve the scientific rigor of XAI research and development, we conduct a comprehensive examination of one of the most commonly used XAI criteria [62]: plausibility. Plausibility assesses the reasonableness of an AI explanation by comparing it with human prior knowledge [38, 62]. Doing so assumes that human explanations provide the ground truth for XAI algorithms. Our critical examination shows that plausibility does not exhibit

Submitted to 39th Conference on Neural Information Processing Systems (NeurIPS 2025). Do not distribute.

construct validity [6, 37] to measure explainability and its key properties, including trustworthiness, understandability, and transparency. In addition, using plausibility as an XAI criterion can pose ethical risks to users by encouraging misleading explanations (explanations that are plausible for wrong AI predictions), which potentially manipulate users and exploit users' trust. To illustrate how plausibility is flawed as an XAI criterion, we provide two Motivating Examples in Box 1.

---

**Box 1**

**Motivating Example 1: XAI for decision support**

Suppose we need to equip a 90% accurate black-box AI model with a post-hoc XAI algorithm to explain AI predictions. If we use plausibility to select the best XAI algorithm (suppose that the candidate XAI algorithms have similar performances on other evaluation criteria), then decision makers (such as doctors) will be more likely to accept an AI prediction when its explanation is more plausible, say the most plausible XAI algorithm will increase doctors' acceptance rate by 5% compared to the second best plausible XAI algorithm.

However, according to the definition of plausibility, since plausible explanations are unconditioned on the correctness of AI predictions, for the increased 5% acceptance cases, they have the same 10% probability of being incorrect as the rest of acceptance cases. Then the rate of misleading cases (doctors' rate of accepting incorrect AI predictions for their plausible explanations) is also increased by $10\% \times 5\% = 0.5\%$. An empirical study of this phenomenon is shown in Appendix E.2. In this scenario, unconditionally making AI explanations more plausible regardless of prediction correctness would be more likely to occlude the signals of incorrect predictions.

**Motivating Example 2: XAI for debugging or bias detection**

Suppose we need to use XAI algorithms to debug AI models, such as detecting biases or wrong patterns learned by AI. If we select or optimize an XAI algorithm based on its plausibility performance, then according to the definition of plausibility, since plausible explanations are unconditioned on the signals of the wrongly learned patterns, unconditionally making AI explanations more plausible would be more likely to occlude the signals of the wrongly learned patterns. This phenomenon is empirically shown in [46].

---

Because using plausibility as an XAI criterion lacks demonstrable benefits, but rather introduces substantial risks of being unscientific and unethical, **we posit that the community should not use plausibility as an XAI criterion to optimize and evaluate the XAI algorithms.** This means that human explanations should not be regarded as the ground truth for XAI. Our analysis also yields the following findings[1]: 1) we point out how to use plausibility properly: plausibility can be used, not as an end, but as a means to facilitate measures of XAI utilities to users, including users' intended purposes of using XAI and human-AI team performance. 2) We identify the proper ways to measure or improve trustworthiness, understandability, and transparency for AI explanations, and 3) identify the mathematical conditions to achieve complementary human-AI performance with XAI. These findings emphasize important yet under-explored research directions that embed users' benefits and perspectives in XAI design and evaluation [94], so that to ensure XAI fulfill its intended function as a critical check and balance mechanism to hold AI systems accountable.

## 2  Alternative view: plausibility as the measure of explainability

We provide a formal definition of plausibility $P$ in the context of XAI:

$$P = \text{similarity}(\mathsf{E}^{\text{human}}, \mathsf{E}^{\text{AI}}) \tag{1}$$

Plausibility $P$ measures the content similarity or agreement between two explanations, $\mathsf{E}^{\text{AI}}$ and $\mathsf{E}^{\text{human}}$, expressed in the same explanation form $\mathcal{E}$. $\mathsf{E}^{\text{AI}}$ is the machine explanation. $\mathsf{E}^{\text{human}}$ is the human explanation based on human prior knowledge on the given task and/or data. Plausibility can be assessed computationally or by human. Human assessment asks users to directly judge the reasonableness of machine explanations quantitatively or qualitatively [40, 75, 81]. Computational assessment approximates the human assessment of $P$ using annotated datasets on human prior knowledge and computational metrics for similarity comparison: human explanations are usually simplified as a set of important features $\mathsf{A}^{\text{human}}$, such as localization masks of important image features in computer vision tasks [73, 59, 93], localization masks of important words in natural language processing tasks [20, 48], important features or concepts with ranking or attribution [45], or a combination of them. $\mathsf{A}^{\text{human}}$ can be from a human-annotated XAI benchmark dataset [73, 59, 93, 20, 48], or generated by another AI model that performs the corresponding localization task. Similarity can be calculated by feature correlation between $\mathsf{A}^{\text{human}}$ and $\mathsf{A}^{\text{AI}}$ [15], or by using metrics on feature overlap, which are commonly used for localization tasks, such as intersection over union

---

[1]This work is mainly relevant to user-oriented XAI algorithms for purposes such as decision support, knowledge discovery, and troubleshooting for AI models. The XAI algorithms include inherently interpretable and post-hoc methods [71].

(IoU) [73, 63, 20] and pointing game rate (hit rate or positive predictive value) [91, 63, 42, 73]. The nuance between human and computational assessments is detailed in Appendix G.

Currently, plausibility is commonly used as a criterion to optimize and evaluate XAI algorithms for more plausible explanations. There is a growing number of human explanation benchmark datasets to evaluate or optimize XAI for plausibility [73, 59, 93, 20, 48]. In a systematic review of 312 original XAI papers that propose a new XAI algorithm in 2014-2020, among the 181 papers that used at least one quantitative evaluation, 34.3% (62/181) used plausibility as the evaluation criterion, and plausibility is the top-chosen evaluation criterion among the twelve criteria surveyed [62]. Plausibility is also one of the main evaluation criteria implemented in the Quantus XAI programming library [32].

The popularity of plausibility in XAI evaluation is the alternative view (opposed to our position) that regards plausibility as a good measure of explainability. Reasons (**Reason 1**-**8**) for the alternative view can be summarized as follows: Plausibility is exactly how we humans gauge the goodness of an explanation from a human explainer [90] (**Reason 1**), and AI explanations are designed the same way to mimic human explanations as the ground truth [73] (**Reason 2**); more plausible explanations indicate better predictions of the AI model [42] (**Reason 3**); and more plausible explanations indicate the AI model learns more effective features as humans [89] (**Reason 4**). Plausible explanations can make AI systems more transparent (**Reason 5**), improve the trustworthiness of an AI system [31] (**Reason 6**), which in turn improve the task performance of human-AI team (**Reason 7**). Plausible explanations are also more understandable (**Reason 8**). Our critical examination in the next section argues against this alternative view, and reveals why these intuitive reasons are surprisingly flawed in supporting the alternative view.

## 3   Why is plausibility surprisingly problematic to measure explainability?

### 3.1   Why is plausibility invalid to measure explainability?

Plausibility is not a measure of explainability because doing so ignores two important facts:

**Fact 1.** *AI predictions are not ideal and can contain errors, uncertainties, biases, shortcuts, unexpected and newly discovered patterns in training.*

**Fact 2.** *The main purposes of explainability include identifying and articulating the ideal and non-ideal signals (e.g. features, patterns) in the AI prediction-making process.*

However, plausibility does not include the case of deviant signals in its definition. Furthermore, evaluating or optimizing XAI for plausibility can encourage more occlusion of the deviant signals, as illustrated in the two Motivating Examples. This is against the intended purposes of explainability and renders plausibility invalid in measuring explainability, according to measurement theory[2] [6, 37]. The measurement invalidity renders the use of plausibility as an XAI criterion unscientific. As we stated in the Introduction, XAI is intended to function as an adversarial mechanism, equipping users with a critical check-and-balance tool to ensure the accountability of AI systems. The role of XAI is similar to the opponent in an adversarial system, such as discriminator in a generative adversarial network, red team in software development, opposition parties in a government, and reviewers in peer review. Then optimizing or evaluating XAI for plausibility is like providing the opponent a strong incentive to construct a spurious positive semblance, which is the least thing we want from an adversarial mechanism. Due to the adversarial nature of XAI, human judgment of *the goodness of an AI explanation* should not be used to measure *the goodness of the explanatory function*. This refutes **Reason 1** for the alternative view.

From the conclusion that plausibility is invalid to measure explainability, which is identical to state that encouraging AI explanation to mirror human explanation is invalid for explainability, one can infer an equivalent proposition that human explanation is not the ground truth for XAI algorithms. This is further supported by epistemic analysis[3] regarding the knowledge source of ground truth for XAI: XAI by definition is to provide reasons for the AI model's prediction process, so the knowledge source of its ground truth is from the model's internal prediction-making process. Faithfulness, another commonly used XAI criterion, is to measure the alignment of XAI algorithm with its ground truth [38]. Humans' decision-making process is independent of the machine prediction process. Therefore, human explanation does not provide the direct grounding of the knowledge source for XAI

---

[2]Validity, together with reliability, are the two basic properties in measurement theory. "Validity refers to the degree to which evidence and theory support the interpretations of test scores for proposed uses of tests." [6]

[3]The detailed epistemic analysis of the ground truth for XAI algorithms is provided in Appendix C.

algorithms, although the content of human and AI explanations can overlap. This refutes **Reason 2** for the alternative view.

We have concluded that plausibility is an invalid measure of explainability, with the equivalent proposition that human explanation is not the ground truth for XAI algorithms. Next, by refuting **Reason 3-8** for the alternative view, we show why this conclusion and its equivalent proposition seem counterintuitive, and what the consequences are if plausibility is used to evaluate or optimize XAI algorithms.

### 3.2 "Plausibility is invalid to measure explainability," i.e. "human explanation is not the ground truth for XAI," why do they seem counterintuitive?

#### 3.2.1 Because assumptions of human explanation do not automatically hold for AI explanation

One reason for the counterintuitiveness of the conclusions in Section 3.1 is: the definition of XAI [22] frames XAI as an anthropomorphic problem [34, 4] that mimics the role and expectation of human explanation to "explain" or present "reasons" to humans. Therefore, it is intuitive for humans to attribute properties and assumptions of human explanation to AI explanation. Human everyday explanation is assumed to be associated with the inquired information about the explainer's internal decision process, detailed in the following key assumptions. Establishing these assumptions is a prerequisite to meet users' expectations of the normal role and functionality of an explanation.

**Assumption 1** (Basic function of explanation). *Explanation is associated with the key rationales and/or evidence used in the explainer's decision process.*

**Assumption 2** (Intended purposes of explanation). *The quality of explanation (i.e., its associated rationale and evidence) is validly associated with the quality of decision.*

The pursuit for more plausible explanations from AI system seems reasonable because it implicitly assumes that the properties and assumptions of human explanation also hold for AI explanation. However, for AI explanation, merely *designing* an XAI algorithm to have the desired properties and assumptions is insufficient to guarantee their realization, unless the XAI algorithm passes rigorous *evalatuion* on its claimed properties and assumptions, as we stated in the Introduction.

Specifically, to make AI explanation fulfill Assumption 1, which is the basic property of any explanation to establish the provided information as explanation and make the internal decision process transparent, the XAI algorithm needs to pass the faithfulness test to validate that the XAI can make the key features and processes in prediction making transparent for the given AI model and task.

Preconditioned by Assumption 1, Assumption 2 enables users to further use the explanation for their intended purposes based on the valid relationship between rationale/evidence and decision. According to the definition of validity in deductive logic[4], there are three possible combinations that precondition a valid relationship between rationale/evidence and decision, and they are visualized as the three quadrants in Fig. 1: a right conclusion with plausible reasons (quadrant I), a wrong conclusion with implausible reasons (quadrant III), and a right conclusion with implausible reasons (quadrant IV). The combination of a *wrong* conclusion with *plausible* reasons (quadrant II) is always logically invalid according to the definition of validity in logic [36]. And we name such a combination of plausible explanations for wrong decisions the *misleading explanations* in the context of XAI. Misleading explanations can cause harms because they violate Assumption 2 of the expected role and function of explanations. For example, a doctor or a judge can be misled by convincingly-generated AI explanations and thus accept AI's wrong recommendations. We further explore the unethical issues of misleading explanations in Section 3.3.1.

By failing to incorporate human assumptions as the preconditions of XAI in its assessment, evaluating or optimizing XAI for plausibility inherently ignores Assumption 2 of the valid interrelation between the quality of explanation and the quality of prediction. Doing so can increase the likelihood of misleading explanations, as demonstrated in the Motivating Example 1 in the Introduction. This effect be illustrated in Fig. 1: without the establishment of valid interrelation between the quality of explanation ($y$ axis) and the quality of prediction ($x$ axis), pursuing plausible explanations will move the overall distribution of explanations in the 2D diagram upward (shown by the red arrow), and the direction of movement is not meaningfully related to the direction of $y$ axis. This creates the

---

[4]The definition of validity in logic is: "A deductive argument is valid when, if its premises are true, its conclusion must be true." [36]

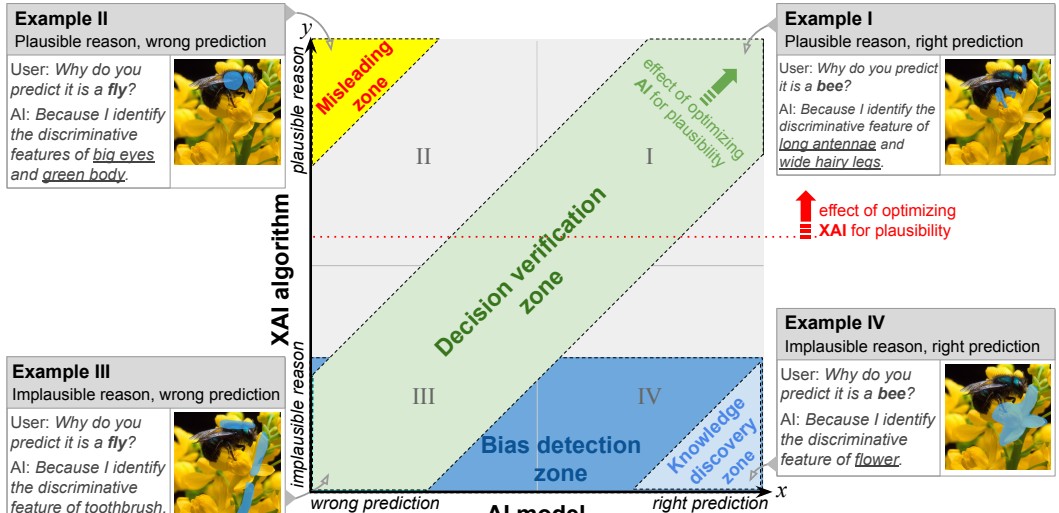

Figure 1: The conceptual 2D distribution diagram of AI explanations from an XAI algorithm regarding the probability of AI decision correctness ($x$ axis) and the degree of plausibility ($y$ axis).

effect of moving more explanations into the misleading zone in quadrant II, indicating the increased likelihood of misleading explanation. Our empirical study also demonstrates this phenomenon that using plausibility as an XAI criterion can increase the likelihood of misleading explanations. The details of the study are in Appendix E.2. Our analysis refutes **Reason 3** for the alternative view, since the interrelation between plausible explanations and good predictions does not hold when using plausibility as an XAI criterion.

### 3.2.2 Because AI *learning* of plausible features is conflated with XAI *presentation* of plausible features

The analysis in Section 3.2.1 indicates that pursuing plausible explanations can become legitimate given that it is preconditioned by necessary human assumptions of explanation. One way to restore the association between plausible explanations and good predictions in Assumption 2 is by incorporating human prior knowledge of important features in AI model training, as seen in previous works in pursuit of "right for the right reasons" [89, 70, 9, 88]. This creates the effect of pushing the distribution of explanations toward the upper right corner in quadrant I in Fig. 1, illustrated by the green arrow. In this process, the optimization of AI models to learn plausible features should be the driving force in order to maintain a valid interrelation between the quality of prediction and the quality of explanation, i.e, the correlation coefficient of $y/x$ should be $\leq 1$ to avoid crossing the misleading zone in quadrant II in Fig. 1. Otherwise, if the optimization of the XAI algorithm for explaining[5] overtakes the optimization of AI model for learning (i.e., the correlation coefficient of $y/x > 1$ and begins to touch quadrant II of the misleading zone), then the speed of increase along the $y$ axis is greater than the speed of increase along the $x$ axis, indicating that the interrelation of the quality of explanation with the quality of prediction is weak. This situation fails to fulfill Assumption 2 and makes the optimization of plausibility illegitimate, and is at risk of increasing the likelihood of misleading explanations.

This analysis indicates that the legitimate optimization of plausibility should make the optimization for AI model's learning of plausible features the driving force, not the optimization for XAI algorithm to present plausible features. This explains the source of confusion: **Reason 4** for the alternative view that "more plausible explanations indicate the AI model learns more effective features as humans" is intuitive and reasonable, because it states the benefit of optimizing plausibility for AI model's learning. But this benefit does not support the conclusion that the XAI algorithm should be selected or optimized for plausibility. Doing so mistakes the target of optimization of AI models for XAI algorithms, and misattributes the improvement in AI model's learning capacity (the cause) to the improvement in XAI algorithm's presentation capability (the effect). Our analysis suggests that when using plausibility to optimize the training of *the AI model to learn* plausible features, it should not be clearly stated as such, and should not be confused with the optimization of *the XAI algorithm to present* plausible features.

---

[5]We note that evaluating and selecting XAI algorithms based on higher plausibility score, which is common for post-hoc XAI algorithms, is also a form of (meta-)optimization of XAI algorithms.

### 3.3 What are the consequences of evaluating or optimizing XAI for plausibility?

In the previous Sections 3.1 and 3.2, we examined the underlying reasons that render plausibility unscientific and unethical as an XAI criterion. Next, we analyze what the consequences are regarding the four main outcomes commonly associated with explainability, as mentioned in **Reason 5-8** for the alternative view. They are transparency, trustworthiness, the task performance of human-AI team, and understandability. For **Reason 5** on the benefits of using plausibility as an XAI criterion for transparency, our analysis of Assumption 1 and previous work [38] show that transparency and plausibility are independent of each other given the AI explanation, and transparency should be measured by faithfulness. We provide a causal diagram showing the conditional independence in Appendix D. Next, we focus on the relationship of plausibility with trustworthiness (**Reason 6**), human-AI team performance (**Reason 7**), and understandability (**Reason 8**).

### 3.3.1 Using plausibility as an XAI criterion can destroy trust and manipulate users

**Reason 6** to enhance stakeholders' trust in an AI model by making XAI algorithms plausible is based on the rationale that [**Premise 1**] a plausible explanation can increase user's *local trust in an AI prediction*, thus [**Premise 2**] the *global trust in the AI model* can be increased accordingly by the accumulation of high local trust in AI predictions. Premise 1 is consistent with our daily experience [58, 52, 57, 90]. Prior empirical studies provide support for Premise 1 [14, 64], and we provide additional empirical evidence on this relationship between user's local trust and explanation plausibility in Appendix E.1.

Premise 2 does not hold because *global* trust is a complex process that cannot be simply reduced to a linear combination of *local* trust. According to trust theories, global trust is developed by first assessing the dependability and reliability (such as credentials, previous records, and reputation) of an entity that provides a partial foundation for provisional global trust [54]. This provisional global trust is then deepened by repeated interactions in three stages of calculus-, knowledge-, and identification-based trust. In the first stage, calculus-based trust is based on the belief that the other will be punished if being untrustworthy. The second stage of knowledge-based trust is grounded in more information that makes the other's behavior more predictable. Predictability enhances global trust even if the other is predictably untrustworthy. Lastly, the third stage of identification-based trust is the belief that one's interests can be fully defended and protected without monitoring. Positive experiences in the interactions can stabilize global trust at a certain stage or move to the next stage. As the stage progresses, trust becomes harder to build as well as to destroy [49].

Regarding our case of XAI, as we emphasized in Facts 1 and 2 in Section 3.1, AI predictions are not ideal, and the role and responsibility of XAI are to faithfully present the deviant-from-ideal signals in the AI prediction process. This indicates that when the certainty or quality of an AI prediction is low, user's *local* trust in the AI prediction should be low to reject AI, and vice versa. In other words, to enhance global trust in the AI system, the role of XAI is not to *enhance* local trust, but to *calibrate* local trust [92] in particular predictions to make the AI model's behavior more predictable to users according to trust theory [49].

As we analyzed in Section 3.1, evaluating or optimizing XAI for plausibility can neither enable XAI to perform its intended role to present non-ideal AI prediction process nor calibrate user's local trust. To the contrary, making XAI algorithms plausible can increase the likelihood of misleading explanations shown in Section 3.2.1. These misleading explanations can manipulate users, take advantage of the fact in Premise 1 and exploit users' trust in AI with specious explanations [54], which eventually can lead to users' distrust.

One may argue that despite the unethical issue of misleading explanations and the possibility of distrust, evaluating or optimizing XAI algorithms for plausibility can still create benefit by improving the task performance of human-AI team. Therefore, in certain circumstances, the benefit may overweigh the drawbacks, which still make plausibility a legitimate criterion for XAI algorithms. We argue against this opinion, since it falsely frames ethics, scientific integrity, and users' autonomy as a trade-off with performance, not as the prerequisites for performance improvement. This trade-off reflects the long-standing tension between explanation and prediction [12, 72]. As an analogy, the relationship between ethics and performance can be regarded as the brake and the engine of a car, similar to the adversarial system we mentioned. Falsely framing them as a trade-off can limit how far the performance can go. Our analysis in the next subsection indicates that there may be a third way to synergize, not to trade off, explanation and performance.

### 3.3.2 Using plausibility as an XAI criterion undermines human autonomy and cannot achieve complementary human-AI performance

We use the accuracy metric to measure task performance. In the context of collaborative human-AI team, suppose $h$, $m$, and $t$ represent the accuracies of human, AI, and human-AI team, respectively, then ideally with the assistance of AI prediction and its explanations, we want the human-AI team to outperform either human or AI alone $t > \max(h, m)$. This is termed complementary human-AI team performance in the literature [8, 92, 50, 40]. Complementary human-AI performance may be regarded as one of the most important utilities of XAI in high-stakes decision-support tasks [8, 86].

It is intuitive to see that by optimizing XAI algorithms for more plausible explanations, it maximizes local trust, and humans would tend to rely more on AI. Provided $m > h$, then the team accuracy $t$ can increase compared to human performing the task alone $h$. However, there is an upper bound of $t$ that cannot exceed $m$, as shown in Theorem 1 (proofs for theorems are in Appendix F). This means complementary human-AI team accuracy cannot be achieved when using plausibility as the XAI criterion.

**Theorem 1** (Case of Impossible Complementarity for XAI). *Let $h$, $m$, and $t$ be the accuracies of the human, AI, and human-AI team, respectively; and $f(P_i)$ be a function of the explanation plausibility $P_i$ denoting the probability of human acceptance of the AI suggestion for the instance $x_i \in \mathcal{D}$, then:*

*If plausibility is independent of the AI decision correctness, then the human-AI team can never achieve complementary accuracy, i.e.: $t \leq \max(h, m)$.*

Theorem 1 is also evidenced by empirical studies [40, 8, 16]. Here, the maximum gain in performance is equivalent to delegating the decision-making task to a black-box AI with accuracy $m$. The involvement of human decision-maker and XAI provides no benefit to task performance over their counterpart black-box model alone. Furthermore, human autonomy is undermined in either case: using a black-box model makes the decision process opacity to inspect and contest, while optimizing XAI algorithms for more plausible explanations increases misleading explanations to deceive users. To summarize, using plausibility as the XAI criterion fails to enable XAI to perform its expected outcomes to improve collaborative human-AI task performance and support human autonomy in decision making.

Given that humans and AI err differently, the ideal role of AI explanation in improving performance is to help humans discern potential uncertainty and mistakes in AI [7], so humans can overwrite AI's potentially uncertain or incorrect predictions with their own judgment. Theorem 2 shows the theoretical conditions on plausibility to achieve complementary human-AI team performance.

**Theorem 2** (Conditions for XAI Complementarity). *Let $h$, $m$, and $t$ be the accuracies of the human, AI, and human-AI team, respectively; $f(P_i)$ be the probability of human acceptance of an AI suggestion for an instance $x_i \in \mathcal{D}$, where $f(.)$ is a monotonically non-decreasing function of the explanation plausibility $P_i$; and $P_i^r$ and $P_i^w$ be the plausibility values of an AI explanation when an instance $x_i$ is predicted correctly or incorrectly, then:*

*Complementary human-AI accuracy can be achieved, i.e., $t > \max(h, m)$, when*

$$\begin{cases} h \geq m & and & \mathbb{E}[f^r] > \frac{h(1-m)}{m(1-h)}\mathbb{E}[f^w]; & or, \\ m > h & and & \mathbb{E}[f^r] > \frac{h(1-m)}{m(1-h)}\mathbb{E}[f^w] + \frac{m-h}{m(1-h)} \end{cases}$$

*where $\mathbb{E}[f^r]$ and $\mathbb{E}[f^w]$ are the expectations of $f(P_i^r)$ and $f(P_i^w)$ over the dataset $\mathcal{D}$, indicating among the correctly or incorrectly predicted instances of AI, how many are accepted by human.*

From Theorem 2, we can get Corollary 1 in Appendix F that $\mathbb{E}[f^r]$ should be greater than $\mathbb{E}[f^w]$, and accordingly, the mean plausibility for correctly predicted data $\mathbb{E}[P^r]$ should be greater than the mean plausibility for incorrectly predicted data $\mathbb{E}[P^w]$ as a prerequisite to fulfill the conditions in Theorem 2. This indicates that plausibility can be evaluated or optimized to correlate with AI decision correctness to achieve complementary human-AI performance. Furthermore, since a low $\mathbb{E}[P^w]$ is preferred and a high $\mathbb{E}[P^w]$ is the definition of misleading explanation, fulfilling conditions in Theorem 2 reduces the number of misleading explanations. It also enables users to appropriately calibrate their local trust depending on prediction reliability [92, 63], and making the model behavior including its potential limitations and mistakes more predictable to users. This in turn may improve global trust, as it is in accordance with the above knowledge-based trust in the trust theory [49, 68]. We conduct a simulation experiment in Appendix E.3 to explore variable interactions and their effect

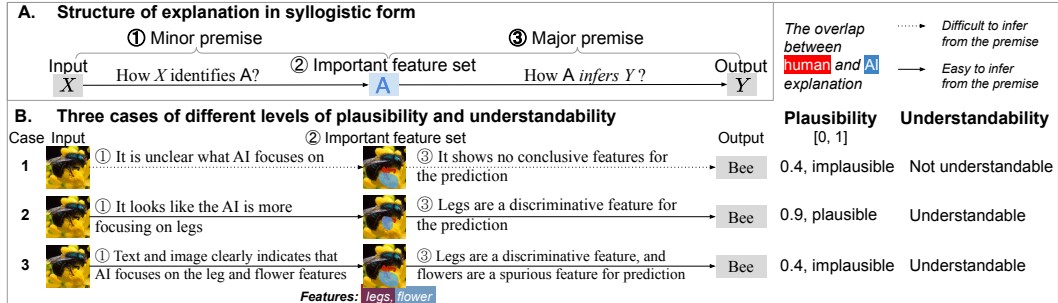

Figure 2: Illustration of the form of explanation and the three cases on understandability.

on team performance, with the empirical results aligning with theoretical findings of Theorems 1 and 2.

### 3.3.3 Plausibility improves understandability at the expense of neglecting other possibilities of enhancing understandability

Lastly, we investigate the relationship between plausibility and understandability. Based on deductive logic and relevance theory in pragmatics [76], we propose a general framework of how humans make sense of the AI prediction process based on the provided explanation. As shown in Fig. 2-A, we dissect the human sense-making process into three steps ①-③ according to the form of syllogism[6]: ① is the minor premise in a syllogism that answers "How is the important feature set A identified from the input $X$?"; ② is the presentation of the important feature set A; and ③ is the major premise that answers "How does the important feature set A infer the prediction $Y$?" An explanation make the prediction process understandable by providing premises and features to facilitate users' successive steps of intuitive inference from input $X$ to conclusion $Y$, i.e.: explanations help users connect dots in their reasoning from $X$ to $Y$ [19, 56, 55].

We argue that plausible features can only contribute a small portion to understandability by corresponding to premises ("dots") that are easy to infer (connect) in user's chain of reasoning (sufficient condition), but cannot provide other forms of information to complete users' chain of reasoning (necessary condition). To illustrate, let's look at three cases in Fig. 2-B:

In this task to classify bees vs. flies from images, we show three explanations with different plausibility scores $P_1$ to $P_3$ and human understandability levels $U_1$ to $U_3$. Explanations are shown in the explanation forms of image mask and text to denote the important feature set A. AI and human explanations are shown in blue and red, respectively.

For Case 1, $U_1$ is low, because the explanation failed to help humans infer meaningful premises: humans cannot identify a definitive feature from the implausible image features, and cannot infer a conclusion from the features accordingly. Here, the implausible features fail to provide relevant information for the given context that facilitates humans' interpretation [76].

Case 2 has improved understandability $U_2$ compared to Case 1, because the human inferential steps are all connected: By having plausible features that are similar to humans' important features, the AI explanation directly leverages the existing human knowledge and inferential steps. The difference between Cases 1 and 2 shows plausible features are a sufficient condition for understandability.

Case 3 shows that without leveraging plausible features, understandability can still be achieved by increasing the expressive power of the explanation form $\mathcal{E}$. Case 3 changes the explanation form from an image mask to a mask with text describing the highlighted features. Although it has the same low plausibility score as Case 2, Case 3 still improves understandability, because the text description provides explicit evidence to confirm the important features that are implicitly indicated in the mask [76]. This indicates that as long as the explanation can strengthen the premises in humans' chain of reasoning from input to prediction, understanding can be reached without being confined to human plausible features. Case 3 provides a counterexample for the argument that "plausible features are a necessary condition for understandability."

The three cases show that plausible features are a sufficient but not necessary condition of understandability. This greatly limits the use of plausibility as a measure of understandability, including: 1) To achieve understandability, the features need to be plausible enough to reduce ambiguity in human inference, otherwise there is still uncertainty in humans' interpretation and understanding. This means in a range from 0 to 1, an increase of plausibility score from 0.1 to 0.4 may not be helpful for understandability, because much ambiguity still remains in the explanation. This phenomenon

---

[6] "A syllogism is a deductive argument in which a conclusion is inferred from two premises." [36]

suggests that plausibility does not have a linear relationship with understandability, and an increment in plausibility may not necessarily lead to an increment in understandability. 2) Because plausible features are only a sufficient but not necessary condition for understandability, explanations with plausible features are a small subset of understandable explanations. There are other kinds of understandable explanations that achieve understandability without being plausible, for example, by strengthening human inferential steps as shown in Case 3. This shows plausibility cannot measure the whole spectrum of understandable explanations. 3) Since plausibility cannot cover the full spectrum of understandable explanations, when using plausibility as the XAI criterion, explanations with plausible features are prioritized over other possibilities of understandable explanations. This may discourage the exploration of other possible approaches to achieve understandability.

## 4    How to use plausibility properly? Use it as a means, not as an end

Our examination in Section 3 shows that although using plausibility as an XAI criterion may seem intuitive, it is actually invalid because doing so violates the prerequisites and assumptions that enable XAI algorithms to perform its intended functions and purposes. This provides implications for XAI evaluation in general: First, the evaluation of XAI should prioritize tests on the fulfillment of Assumption 1 that establishes the piece of information as "explanation" and aims to make the AI prediction process transparent. Such tests are termed faithfulness in XAI evaluation [38], which should be considered as the basic test for any XAI algorithm. Then depending on specific usage scenarios, optional evaluations of XAI can be conducted to assess the fulfillment of the particular intended purposes of using AI explanations and their underpinning assumptions.

For example, Assumption 2 underpins several primary intended purposes, including decision verification/trust calibration, bias and bugs detection, new knowledge discovery. Different purposes have different required correlations between the quality of explanation and the quality of prediction, which can be visualized as the corresponding zones in Fig. 1. Decision verification/trust calibration is further associated with the downstream objective of the task performance of human-AI team. For these intended purposes, user studies and computational assessments can be performed to measure how well the quality of explanation can exhibit the desired interrelation with the intended purposes.

Because the quality of explanation defines plausibility, plausibility measure can play a role in the assessment of these intended purposes of XAI. Here, plausibility is no longer the end objective of XAI evaluation and optimization. It is an intermediate measure to facilitate the downstream assessment on the interrelation between plausibility and the quality of prediction. We list prior works as examples that use plausibility as a means for the intended purposes of decision verification [42] and bias detection [2, 78].

In addition to the evaluation on the efficacy of XAI algorithms, another equally important evaluation aspect is conducting thorough assessments of the scopes, limitations, weaknesses, failure modes, and risks of XAI algorithms [43, 13, 44]. Our examination identifies the unethical issue of misleading explanations. Controlling its number under a certain threshold and declaring its probability of occurrence and potential risks should be considered as an important aspect of the evaluation and limitation acknowledgment of XAI algorithms.

## 5    Conclusion

To improve the scientific rigor of XAI, we conduct a critical examination of the use of plausibility as an XAI criterion. Our examination shows using plausibility as the XAI criterion is unscientific, because plausibility could not measure explainability, transparency, and trustworthiness, and cannot measure the full spectrum of understandability. Using plausibility as the XAI criterion is also unethical, because it increases misleading explanations, can cause distrust, and is detrimental to human autonomy. Therefore, we call the community to stop using plausibility as the XAI criterion to evaluate or optimize XAI algorithms. This means human explanations are not the ground truth for XAI algorithms.

Our analysis also suggests ways to improve XAI: Transparency can be improved by increasing faithfulness. Understandability can be improved by increasing the expressive power of the explanation form. Trustworthiness and human-AI team performance can be improved by enabling users to appropriately calibrate their local trust, and we provide two theorems that identify the mathematical conditions to achieve complementary human-AI performance. We emphasize that the optimization of AI model to learn plausible features should not be confused with the optimization of XAI algorithms to present plausible features. We also suggest ways to improve XAI evaluation paradigm by using plausibility as an intermediate measure to optimize users' intended purposes of using AI explanations.

## Impact Statement

By critically examining the common criterion of explainable AI, this work aims to prevent the negative impacts of explainable AI techniques if optimized or evaluated inappropriately. Contrary to the common sense that developing and deploying ethical AI techniques —- such as explainable AI —- can always create positive societal impacts, we argued in the beginning of this paper that if explainable AI techniques are not properly developed and assessed, they could create the "ethics washing" effect [85, 25, 67, 27] that causes harms by "making unsubstantiated or misleading claims about, or implementing superficial measures in favour of, the ethical values and benefits of digital processes, products, services, or other solutions in order to appear more digitally ethical than one is." [25]

From our critical examination, we identified the negative societal impacts of using plausibility as the criterion to evaluate or optimize explainable AI algorithms, including: increasing the likelihood of misleading explanations that can deceive and manipulate users to trust or accept faulty AI suggestions; undermining human autonomy; being detrimental to the task performance of the human-AI team; and influencing the research agenda of explainable AI by ignoring other possibilities of enhancing understandability. We hope this work can facilitate the community's critical inspections of current practice in the research and development of explainable AI to achieve its intended ethical purposes and create more positive societal impacts.

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

# Appendix

# Table of Contents

# A Definition of relevant XAI terms

Table 1: As there is a lack of unified definitions for the key concepts commonly encountered in the XAI field, and some concepts are often intertwined with each other, we provide the working definitions to clarify the scope of the concepts discussed in this work.

| Term | Definition |
| --- | --- |
| **Accountability** | According to Doshi-Velez et al. [23], accountability is "the ability to determine whether a decision was made in accordance with procedural and substantive standards and to hold someone responsible if those standards are not met." |
| **Explainability** | In this work, we use explainability and interpretability interchangeably to denote the feature in an AI system that can explain the rationales of AI decisions to users in understandable ways. Explainability/interpretability differs from AI model visualization in that explainability emphasizes the intention and behavior of "explaining" and complies with all the social assumptions in human explanatory communication [30, 76]. |
| **Faithfulness** | Faithfulness is the level at which explanations accurately represent the prediction process of the AI model [38]. In the literature, it is also called fidelity or truthfulness [28, 42]. |
| **Plausibility** | According to Jacovi and Goldberg [38], plausibility is how convincing the explanation is to humans, and they differentiated plausibility from faithfulness, where the former relied on human judgment or human-provided explanations involved. |
| **Transparency** | According to Markus et al. [53], a model is transparent "if the inner workings of the model are visible and one can understand how inputs are mathematically mapped to outputs." While some XAI literature uses transparency as a synonym for understandability [10, 5], we use transparency to emphasize exposing the inner workings of the AI model, and use understandability to emphasize the human factors in comprehending the decision rationales of an AI model. |
| **Trustworthiness** | According to the Oxford English Dictionary, trustworthiness is "the ability to be relied on as honest or truthful." |
| **Understandability** | According to Barredo Arrieta et al. [10], understandability "(or equivalently, intelligibility) denotes the characteristic of a model to make a human understand its function – how the model works – without any need for explaining its internal structure or the algorithmic means by which the model processes data internally." |

 **B   Symbol table**

Table 2: Reference of symbols and their definitions used in the paper. r.v. – random variable

| Symbol | Definition | Introducing place |
|---|---|---|
| $\mathsf{A}$ | Important feature set | Section 2 |
| $\mathsf{E}$ | Explanation | Eq. (1) |
| $\mathcal{E}$ | Explanation form | Section 2 |
| $\mathsf{Pr}$ | Probability | Theorem 1 |
| $x_i$ | An data instance in the dataset $\mathcal{D} = \{x_1, \ldots, x_N\}$ | Theorem 2 |
| $P$ | A real-valued r.v. of the plausibility measure of an explanation $\mathsf{E}$. Its subscript $i$ denotes $P$ is for the explanation of an instance $x_i \in \mathcal{D}$ | Eq. (1) |
| $C_i$ | A Bernoulli r.v. of an instance $x_i$ being correctly predicted by a decision-maker. The superscript of $C$ denotes the identity of the decision-maker being machine, human, or team (human assisted by AI). | Proof of Theorem 1 |
| $h$ | Accuracy of human performing the task alone on the given dataset $\mathcal{D}$ | Theorem 1 |
| $m$ | Accuracy of an AI model performing the task alone on the given dataset $\mathcal{D}$ | Theorem 1 |
| $t$ | Accuracy of the human-AI team performing the task on the given dataset $\mathcal{D}$ | Theorem 1 |
| $B_i$ | A Bernoulli r.v. of the AI suggestion of instance $x_i$ being accepted by humans | Proof of Theorem 1 |
| $f(P_i)$ | $\mathsf{Pr}(B_i = 1) = f(P_i)$, parameter for the r.v. $B_i$, denoting the probability of human accepting the AI suggestion for $x_i$ explained by AI explanation $\mathsf{E}_i$ with plausibility value of $P_i$; $f(.)$ is the function of human factors that decide to accept or reject AI given $P_i$ | Theorem 1 |
| $P_i^r$ | Shorthand for $P_i\|C_i^{\mathrm{AI}} = 1$, which is the plausibility $P_i$ of an explanation given the instance $x_i$ is correctly predicted by AI ($C_i^{\mathrm{AI}} = 1$) | Theorem 2 |
| $P_i^w$ | Shorthand for $P_i\|C_i^{\mathrm{AI}} = 0$, which is the plausibility $P_i$ of an explanation given the instance $x_i$ is incorrectly predicted by AI ($C_i^{\mathrm{AI}} = 0$) | Theorem 2 |
| $f(P_i^r)$ | Shorthand for $f(P_i\|C_i^{\mathrm{AI}} = 1)$, which is the human acceptance of an AI suggestion (if it is correctly predicted, $C_i^{\mathrm{AI}} = 1$) explained by AI explanation $\mathsf{E}_i$ with plausibility $P_i^r$ | Theorem 2 |
| $f(P_i^w)$ | Shorthand for $f(P_i\|C_i^{\mathrm{AI}} = 0)$, which is the human acceptance of an AI suggestion (if it is incorrectly predicted, $C_i^{\mathrm{AI}} = 0$) explained by AI explanation $\mathsf{E}_i$ with plausibility $P_i^w$ | Theorem 2 |
| $\mathbb{E}[f^r]$ | Shorthand for $\mathbb{E}[f(P^r)]$, which is the conditional expectations of $f(P_i\|C_i^{\mathrm{AI}} = 1)$. It measures the true positive rate that among the correctly predicted instances of AI, how many are accepted by human | Theorem 2 |
| $\mathbb{E}[f^w]$ | Shorthand for $\mathbb{E}[f(P^w)]$, which is the expectation of $f(P_i\|C_i^{\mathrm{AI}} = 0)$. It measures the false positive rate that among the incorrectly predicted instances of AI, how many are accepted by human | Theorem 2 |
| $\mathbb{E}[P^r]$ | The expectation of $P_i\|C_i^{\mathrm{AI}} = 1$, which is the mean plausibility for correctly predicted data | Corollary 1 |
| $\mathbb{E}[P^w]$ | The expectation of $P_i\|C_i^{\mathrm{AI}} = 0$, which is the mean plausibility for incorrectly predicted data | Corollary 1 |
| $\mathcal{L}$ | The line that depicts the relationship between $\mathbb{E}[f^w]$ and $\mathbb{E}[f^r]$ in Theorem 2 | Eq. (2) |

 **C   Epistemic analysis of the ground truth for XAI algorithms**

We extend the epistemic analysis of the ground truth for XAI algorithms in Section 3.1.  XAI algorithms are grounded in the AI model's internal prediction-making process. AI model's prediction process is grounded in the training data and human prior knowledge. Then can we say that XAI

algorithms are also grounded in the training data and human prior knowledge? This is equivalent to state that the human prior knowledge or human explanations can be served as another ground truth for XAI algorithms, in addition to the ground truth of AI model's prediction process.

We argue that the above two statements are wrong: XAI algorithms are not grounded in the training data and human prior knowledge, and human prior knowledge or human explanations cannot be served as the ground truth for XAI algorithms. This is because in the above deduction from grounding XAI algorithms in the AI model to human prior knowledge, it utilizes the assumption that the training data and human prior knowledge can be reduced to the trained AI model. This assumption does not hold because the training data and human prior knowledge is irreducible to the AI model, according to the view of complexity science [80, 18]. As the common aphorism "All models are wrong, but some are useful" states, AI models can be useful abstractions of the complexities of the training data and human prior knowledge, but cannot fully represent them. There will always be discrepancies between the AI model and training data/human knowledge, such that the performance of AI models cannot reach the perfect state of being error-free.

The fact that human explanations are not the ground truth for XAI algorithms indicates that we should not mistake the explanatory task for a predictive task: in a predictive task, the goal is to predict the most likely human explanation for the given circumstance. However, this is not the goal for XAI algorithms.

# D Plausibility is conditionally independent of transparency

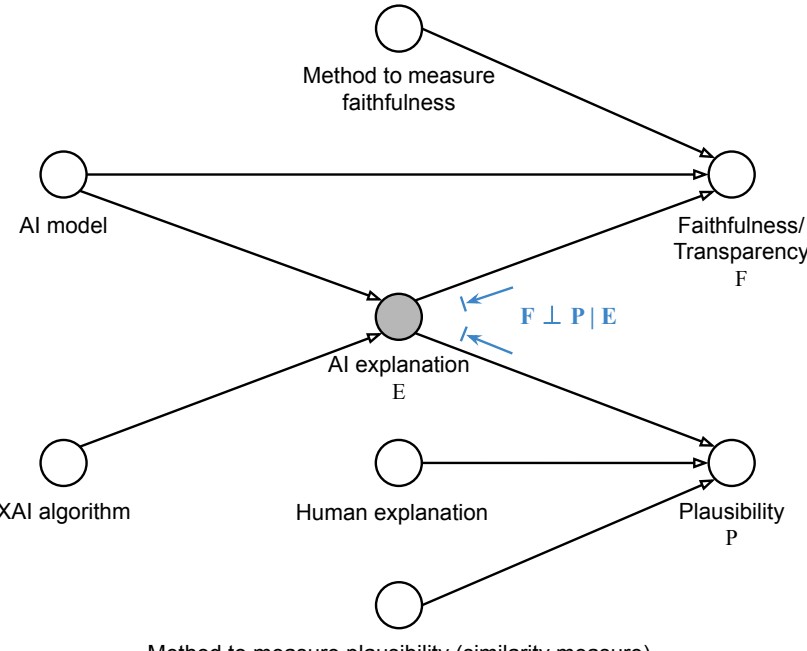

Figure 3: The causal diagram (directed acyclic graph) shows our qualitative causal assumptions on variables related to plausibility and faithfulness/transparency.

In the scope of XAI research, a model is transparent "if the inner workings of the model are visible and one can understand how inputs are mathematically mapped to outputs" [53]. While some literature uses transparency and understandability interchangeably [10, 5], we distinguish the two by using transparency to denote the system aspect of its inner workings, and understandability to denote the human aspect of understanding a model (Table 1). We argue that transparency and plausibility are conditionally independent given AI explanations. Thus, transparency cannot be measured by plausibility.

Once we separate the human aspect from transparency, transparency solely denotes manifesting the inner workings of the AI model's prediction process. The way to manifest is through XAI algorithms.

The capability of XAI algorithms to faithfully manifest the AI model's inner workings defines and is quantified by faithfulness. Then, in the context of XAI, we can use faithfulness as a synonym for transparency. Prior literature has argued that faithfulness and plausibility may be two orthogonal concepts of XAI, and should not be confused with each other [38, 69, 62]. We analyze the relationship between plausibility and transparency/faithfulness using a causal diagram [65, 24, 79] (Fig. 3).

The causal diagram in Fig. 3 represents our causal assumptions. The inputs and the associated method to calculate a variable are considered to be the cause of the variable. AI model and XAI algorithms are plug-ins to each other, and they are assumed to be independent. AI explanation is calculated from the AI model and XAI algorithm. The faithfulness score of an XAI algorithm is calculated using the AI explanations and AI model as inputs to a faithfulness method. The plausibility score is calculated by comparing human and AI explanations using a similarity measure (Eq. (1)). Based on the causal diagram, conditioning on AI explanations $d$-separates[7] plausibility and faithfulness/transparency. In the calculation of faithfulness and plausibility, the AI explanation is an observed variable, then plausibility and faithfulness/transparency are conditionally independent of each other given the AI explanation. Because of the conditional independence between the two, plausibility cannot serve as an indicator to measure transparency.

# E  Empirical studies

We conduct three empirical data analyses:

1. We use data from a doctor user study [40] to test Premise 1 used in  Section 3.3.1 that plausible explanations can increase user's local trust in an AI prediction.

2. We use data from a computational study [42] to demostrate the phenomenon identified in Section 3.2.1 that using plausibility as the criterion to select XAI algorithms can increase the likelihood of misleading explanations.

3. We conduct a simulation experiment to explore the conditions of complementary performance in Section 3.3.2.

The first two data analyses are for new research questions that were not covered by the scope of the original studies. In our data analyses, we use a significance level of $\alpha = 0.05$. Statistical analyses were performed using the Python statistical package Pingouin[8]. Data and code for the analyses are provided in the supplementary material. The data analyses were conducted on a 4-core CPU laptop computer, and the time of execution for the scripts was usually within seconds.

## E.1  Testing the hypothesis on the relationship between plausibility and local trust

**Hypothesis.** In Section 3.3.1 on the relationship between trust and plausibility, we introduce Premise 1 that users have higher local trust in AI suggestions with more plausible explanations. This hypothesis is included in Assumption 2 of human explanation in Section 3.2.1, and is also one of the assumptions (Conjecture 1) for Theorem 2. Here, we aim to test the hypothesis empirically.

**Data.** To test the hypothesis, we conduct a secondary data analysis based on data collected from a clinical user study [40]. The study was conducted in an AI and explanation-assisted clinical decision-making setting. The study recruited 35 neurosurgeons, each reading 25 magnetic resonance images (MRIs) to grade the brain tumor into high or low grade. For each MRI, doctors first gave their initial judgment. Then the AI model provided a second opinion accompanied by its explanation to assist doctors in making a final decision. The explanation was a heatmap showing the important image regions for the AI prediction. Doctors initial judgment and final decision were recorded. Doctors also gave a plausibility score for each AI explanation on a 0–10 scale on the question: "How closely does the highlighted area of the color map match with your clinical judgment?" The study design and results are detailed in [40]. The secondary analysis of data was approved by Anonymous University Research Ethics Board with Ethics Application Number 30001984.

**Variables.** In our analysis, the independent variable is the doctors' plausibility assessment on a scale of 0-10, and the dependent variable is the binary variable of the agreement of doctors' final decisions

---

[7]For the definition of $d$-separation, see Definition 1 in [65].

[8]http://pingouin-stats.org/index.html

with AI predictions. We use humans' behavior of reliance on AI (accept or reject AI suggestion) as an observable variable for the latent variable of human local trust [83, 47]. The variable of doctors' agreements with AI is a weak indicator of doctors accepting AI suggestions, because this task was a binary classification problem, and if doctors final decisions agreed with AI suggestions, it could be due to doctors following their own judgments, or following AI suggestions; If doctors final decisions disagreed with AI suggestions, it was due to doctors following their own judgments and rejecting AI suggestions. Therefore, the group of doctors' disagreement with AI reflects doctors' decisions to reject AI suggestions; the group of doctors' agreements with AI is a mixture of doctors' decisions to accept and reject AI suggestions, and the study design could not differentiate between the two scenarios. Although being imperfect, to the best of our knowledge, this is the only data that provides plausibility measure and approximated behavior measure on trust, and we could not find other publicly available datasets that include these two pieces of information. Future user studies should improve the study design, for example, by directly asking users about their decisions on whether to accept or reject AI suggestions.

**Data distribution.** Table 3 and Fig. 4 show the plausibility distribution for the two groups when doctors agree or disagree with AI.

**Statistical test.** We test the null hypothesis that when doctors agree with AI, the plausibility level is no higher than the plausibility level when doctors disagree with AI. Since the data do not meet the assumption of normality for the $t$-test, we conduct a one-sided Mann–Whitney $U$ test to test the hypothesis. It shows the explanation plausibility score is significantly higher for the group when doctors agree with AI (M±SD: $6.45 \pm 2.82$) than the group when doctors disagree with AI (M±SD: $3.82 \pm 2.56$), $U = 46533.0$, $p$-value= $3.29 \times 10^{-16}$.

Table 3: Statistical summary of physicians' assessment of explanation plausibility for two groups on whether doctors' final decisions agree or disagree with AI suggestions. It lists the mean, standard deviation, min, median, max, 25%, and 75% quantile of the plausibility score on a 0–10 scale.

| Group | Number of decisions | Plausibility | | | | | |
|---|---|---|---|---|---|---|---|
| | | M±SD | Min | 25% | Median | 75% | Max |
| Agree | 649 | 6.45±2.82 | 0 | 5 | 7 | 9 | 10 |
| Disagree | 95 | 3.82±2.56 | 0 | 2 | 4 | 5.5 | 10 |

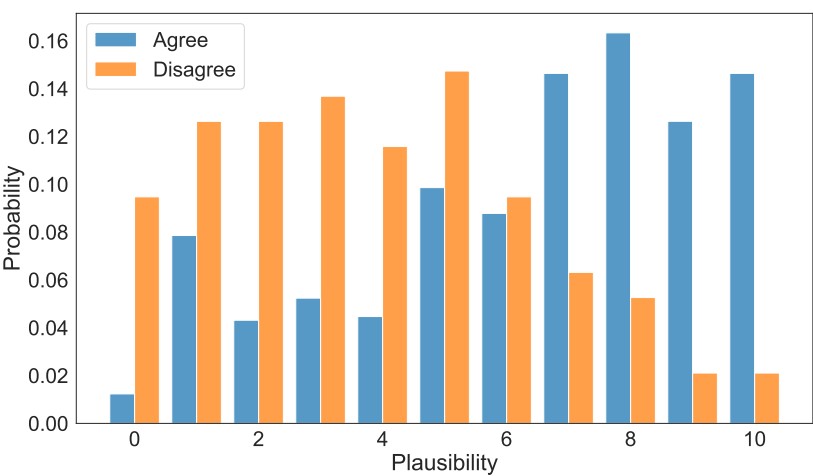

Figure 4: Histogram of physicians' assessment of explanation plausibility on a 0–10 scale. The blue (left) and orange (right) bars are the distributions of groups when doctors' final decisions agree or disagree with AI suggestions, respectively. Since the numbers of data are imbalanced between the two groups, the histograms visualize the relative probability of each plausibility score within a group.

**Causal analysis.** To further determine the causal effect of plausibility on doctors' local trust, we conduct a causal analysis [84]. We calculate the average treatment effect (ATE) [61] of plausible explanation ($X = 1$, which is defined by $P > 5$) to doctors' agreement with AI ($Y = 1$), by controlling the covariant on MRI case easiness $Z$, which is calculated by participants' mean accuracy of each MRI case. Using logistic regression adjustment as an outcome model for $\Pr(Y = 1 | X, Z)$, $\text{ATE} = \Pr(Y^{X=1} = 1) - \Pr(Y^{X=0} = 1) = 0.94 - 0.77 = 0.17$, indicating plausible explanations have an effect of increasing doctor's agreement with AI with a probability of $0.17$.

The above analyses show that AI explanations with higher plausibility leads to doctors' higher agreement with AI suggestions. As stated above, because disagreements can reflect doctors' rejection of AI suggestions, and agreement is a mixture of doctors' acceptance and rejection of AI suggestions, if rejections in the agreement group follow the same distribution as rejections in the disagreement group, then the results using agreement measure tend to underestimate the difference between acceptance and rejection. Therefore, the empirical analyses provide indirect evidence to support our hypothesis that plausible explanations can increase users' local trust manifested in their behavior of being more likely to accept AI decisions.

## E.2 Visualizing the effect of using plausibility for XAI evaluation

**Hypothesis.** In Section 3.2.1 and the Motivating Example 1 in Section 1, we deduce the conclusion that selecting XAI algorithms based on plausibility would increase the likelihood of misleading explanations. This is the hypothesis that we test here using empirical data.

**AI models and XAI algorithms.** This analysis uses data from a computational evaluation of XAI, where five convolutional neural network (CNN) models were trained on a binary medical image classification task to grade brain tumors from MRI images. The five 3D CNN models only differed in their random initialization of parameters. The mean accuracy of the five models was 0.8946±0.0199. Then 16 post-hoc XAI algorithms were used to explain the trained models. The included XAI algorithms use gradient- or perturbation-based methods. The generated AI explanations are in the explanation form of a heatmap that highlights the important regions for prediction. The study details are in [42].

**Variables.** The dataset used to train the AI model includes human-annotated segmentation masks for brain tumors. Therefore, the independent variable of plausibility is calculated by the percentage of important features within the lesion mask. The dependent variable is the number of misleading explanations. A misleading explanation for a data instance $x_i$ is defined as an explanation that has a high plausibility $P_i$ and a low probability of AI prediction correctness $\Pr(C_i^{\text{AI}} = 1)$, and their difference is big enough with $P_i - \Pr(C_i^{\text{AI}} = 1) > \beta$. Both $P_i$ and $\Pr(C^{\text{AI}_i} = 1)$ are in the range of $[0, 1]$, and $\beta$ is set to be a number around the higher tail in a distribution. In our analysis, we set $\beta = 0.75$. We use the probability of the ground truth label to represent $\Pr(C_i^{\text{AI}} = 1)$. We use the average plausibility from a total of 370 test instances to rank $P_i$ of an XAI algorithm. The test instances were aggregated from the five similarly trained models on a test set containing 74 instances.

**Statistical test.** Since the data fail to meet the assumption of normality for Pearson's correlation, we conduct a nonparametric Spearman's correlation test to test the hypothesis. The result shows that among the 16 post-hoc XAI algorithms, there is a high Spearman's correlation between the percentage of misleading explanations and the average plausibility, $r(14) = 0.84$, $p$-value$= 4.7 \times 10^{-5}$. Fig. 1 visualizes the distribution of misleading explanations of the 16 XAI algorithms. This observation is in accordance with our conclusion in Section 3.2.1 that evaluating or selecting XAI based on high plausibility increases the likelihood of misleading explanations.

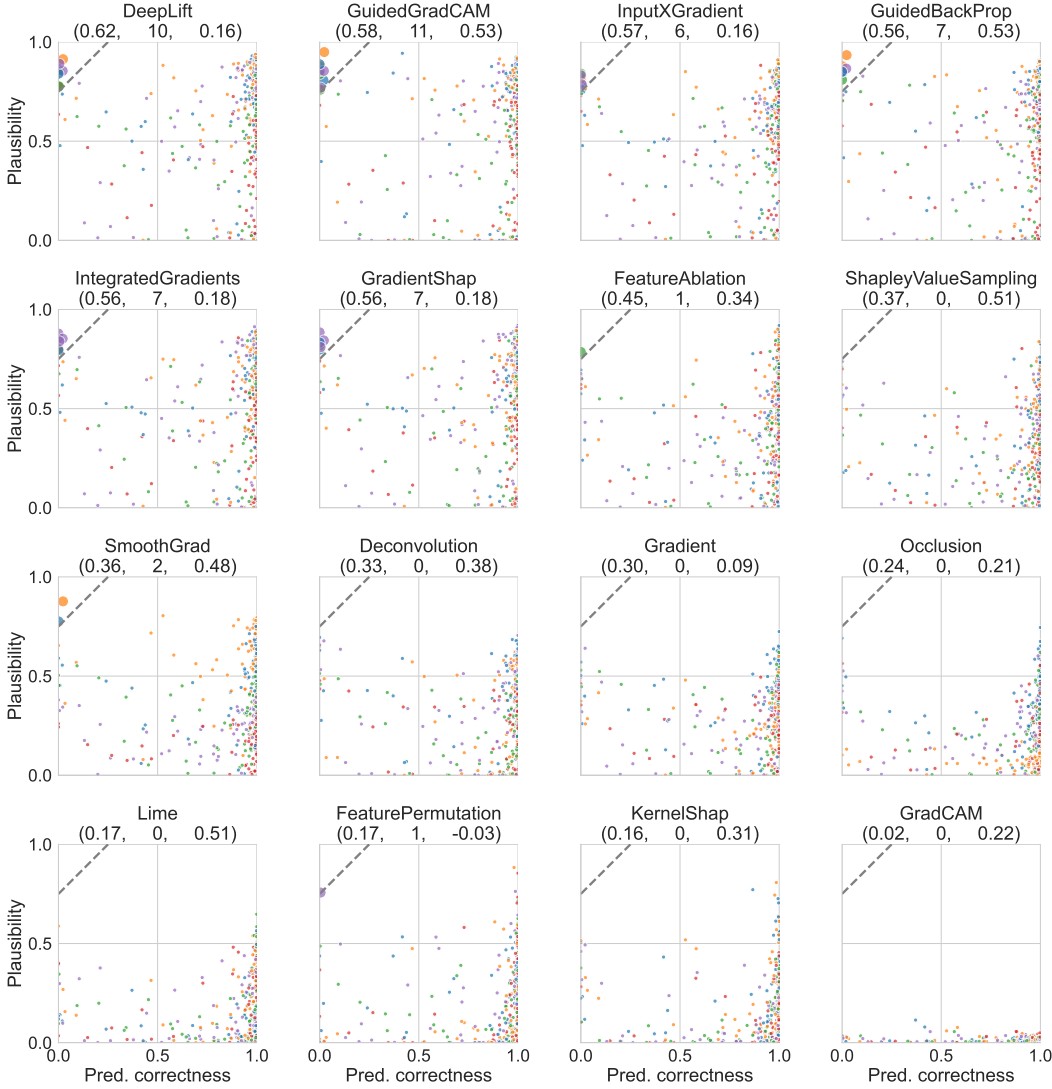

Figure 5: The 2D distribution of AI explanations regarding the probability of AI decision correctness $\Pr(C^{\mathrm{AI}} = 1)$ ($x$ axis) and plausibility $P$ ($y$ axis) for the 16 XAI algorithms. Each dot is a test instance, and the color represents the identity of the five similarly trained models. Each subplot is the conceptual plot of Fig. 1 populated by empirical data. The misleading zone is the upper left corner ($P - \Pr(C^{\mathrm{AI}} = 1) > 0.75$) indicated by a dashed line. The dot size for misleading explanations is enlarged for better visibility. The order of subplots is ranked by the mean plausibility of XAI algorithms. The three numbers under the name of an XAI algorithm are: mean plausibility, number of misleading explanations out of the total 370 instances, and mean faithfulness (measured by gradual feature removal in [42]) of an XAI algorithm on the five models.

**Limitation.** A limitation of this analysis is that the conclusion is drawn from XAI algorithms with different levels of faithfulness to the given model and task. As we proposed in Section 4, faithfulness is the basic evaluation for XAI algorithms. Therefore, ideally this analysis should be accompanied by the analysis on XAI algorithms that achieve a certain threshold of faithfulness. We can deduce that faithfulness may not influence the results, because faithfulness is conditionally independent of plausibility according to the conclusion of Appendix D. However, it would still be beneficial to conduct empirical studies to validate it. From Fig. 5 we can see that among the five XAI algorithms with the same level of higher faithfulness ($0.48 \sim 0.53$ of Guided GradCAM, Guided Backprop, Shapley Value Sampling, SmoothGrad, and LIME), selecting XAI algorithms based on higher plausibility still has the same tendency to increase the likelihood of misleading explanations. However, the sample size here in our experiment is too small to conduct statistical tests, and future experiments are needed to test the hypothesis that given the same satisfactory level of faithfulness, selecting XAI algorithms for high plausibility can increase the number and misleading explanations.

### E.3  Simulation experiment on human-AI collaboration and complementary performance

**Experiment setup.** We conduct simulation experiments of human-AI collaboration to study the factors of plausibility, human and AI performance, and their relationship to complementary performance in Theorem 2. In a human-AI collaborative setting, the experiment simulates the ground truth labels, human and AI predictions, the plausibility score, and the human acceptance of AI prediction in a classification problem. We generate the explanation plausibility score $P$ from a normal distribution with the mean randomly drawn in the range of $[0, 3)$ when the AI prediction is correct, and in the range of $[-3, 0)$ when the AI prediction is incorrect (i.e., plausibility values reflect correctness). We set the human factor function of accepting an AI prediction $f(P)$ to be the sigmoid function of $P$. Then the team prediction is the AI prediction if the human accepts AI or the human prediction otherwise. From the team predictions, we can calculate the team accuracy, $\mathbb{E}[f^r]$ and $\mathbb{E}[f^w]$, and conclude if complementary accuracy is achieved or not. Each simulation trial is run on 2000 test data instances in a 10-class classification task. Some data samples generated from the scripts of the simulation experiment are shown in Table 4.

Table 4: Ten data samples showing how data are generated from the scripts of the simulation experiment. In a five-class classification task, we generate the ground truth (GT) labels. Then human and AI predictions for each data instance are generated according to their preset accuracies. In this data sample, the human accuracy is 0.7, AI accuracy is 0.9. Then plausibility score $P$ is generated based on the information of AI correctness. We use the sigmoid function for $f(P)$ of the human likelihood of accepting an AI prediction. The human-AI team prediction is the AI prediction if the human accepts an AI prediction, otherwise it is the human prediction. Then the human-AI team accuracy can be calculated from the team prediction. In this case, the team accuracy is 1.0, which achieves complementary accuracy.

| Data ID | Human prediction | AI prediction | $P$ | $f(P)$ | Accept AI | Team prediction | GT |
|---|---|---|---|---|---|---|---|
| 01 | 5 | 5 | 2.47 | 0.92 | True | 5 | 5 |
| 02 | 2 | 2 | 1.14 | 0.76 | True | 2 | 2 |
| 03 | 5 | 5 | 0.79 | 0.69 | True | 5 | 5 |
| 04 | 3 | 3 | 3.75 | 0.98 | True | 3 | 3 |
| 05 | 3 | 4 | 1.46 | 0.81 | True | 4 | 4 |
| 06 | 1 | 2 | 1.13 | 0.76 | True | 2 | 2 |
| 07 | 2 | 5 | 1.76 | 0.85 | True | 5 | 5 |
| 08 | 3 | 1 | -1.88 | 0.13 | False | 3 | 3 |
| 09 | 2 | 2 | 1.90 | 0.87 | True | 2 | 2 |
| 10 | 1 | 1 | 0.74 | 0.68 | True | 1 | 1 |

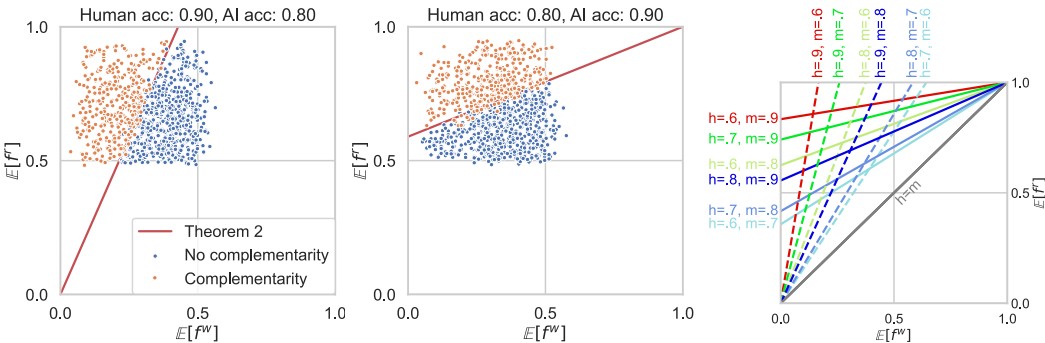

Figure 6: Visualization of the simulation experiment. Left and middle panels: Results of the simulation experiment on the $\mathbb{E}[f^w]$-$\mathbb{E}[f^r]$ plot. Each panel has 1000 dots, which represents 1000 simulation trials. Orange and blue dots indicate whether complementary performance is achieved or not in a trial, respectively. The $\mathcal{L}$ line in Eq. (2) is shown in red to visualize the relationship between $\mathbb{E}[f^w]$ and $\mathbb{E}[f^r]$ in Theorem 2. The left and middle panels show two conditions when human accuracy is greater or less than AI accuracy. Right panel: The relationship of human and AI accuracies $h$ and $m$ in the $\mathbb{E}[f^w]$-$\mathbb{E}[f^r]$ plot according to the formulas in Theorem 2. Each line is the $\mathbb{E}[f^w]$-$\mathbb{E}[f^r]$ line according to different values of $h$ and $m$.

**Theorem 2-related results.** We show the results with respect to the relationship between $\mathbb{E}[f^w]$ and $\mathbb{E}[f^r]$ in Fig. 6 left and middle panels. We define line $\mathcal{L}$ (red lines in Fig. 6 left and middle panels) as the line with the same slop and intercept between $\mathbb{E}[f^w]$ and $\mathbb{E}[f^r]$ as depicted in Theorem 2.

$$
\mathcal{L}: \begin{cases} \mathbb{E}[f^r] = \frac{h(1-m)}{m(1-h)}\mathbb{E}[f^w] & \text{if } h \geq m \\ \mathbb{E}[f^r] = \frac{h(1-m)}{m(1-h)}\mathbb{E}[f^w] + \frac{m-h}{m(1-h)} & \text{if } m > h \end{cases} \tag{2}
$$

The plots show that the simulation experiment confirms the theoretical finding in Theorem 2 that trials achieved complementary accuracy (the orange dots) reside above the line $\mathcal{L}$, which correspond to the solution space where the relation between $\mathbb{E}[f^w]$ and $\mathbb{E}[f^r]$ in Theorem 2 holds. The two plots show that it is possible to achieve complementary accuracy when human accuracy is either greater or less than AI accuracy (Corollary 3 in Appendix F). The two plots also show that the two $\mathcal{L}$ lines are symmetric around the diagonal $\mathbb{E}[f^r] = 1 - \mathbb{E}[f^w]$. We further illustrate the relationship of different values of human and AI accuracies $h$ and $m$ in Fig. 6-right that confirms this symmetric relationship. The plot shows that as the values of $h$ and $m$ become closer to each other, the possibility of achieving complementary accuracy gets higher as the area above the $\mathcal{L}$ line grows bigger. We illustrate this relationship with more value assignments of $h$ and $m$ in Fig. 7 and Fig. 8. The $\mathcal{L}$ line always resides on or above the $\mathbb{E}[f^r] = \mathbb{E}[f^w]$ diagonal towards the upper left corner, when $\mathbb{E}[f^r]$ is larger and $\mathbb{E}[f^w]$ is smaller. This confirms Corollary 1 in Appendix F that $\mathbb{E}[f^r]$ is always greater than $\mathbb{E}[f^w]$ when complementary accuracy is potentially achievable. This also indicates that if plausibility distribution can enable users to reliably know when to accept AI and when not to, the distribution of the human-AI collaboration experiment result (the dots) will more likely reside above the line $\mathcal{L}$ and are more likely to achieve complementary accuracy.

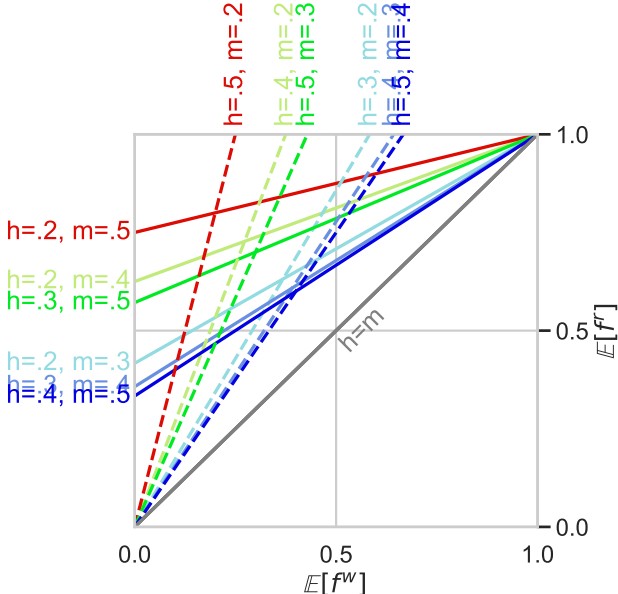

Figure 7: The relationship of human and AI accuracies $h$ and $m$ in the $\mathbb{E}[f^w]$-$\mathbb{E}[f^r]$ plot according to the formulas in Theorem 2. Each line is the $\mathbb{E}[f^w]$-$\mathbb{E}[f^r]$ line according to different values of $h$ and $m$. In Fig. 6-right, we show a similar plot when $h$ and $m$ are above $0.5$. This plot shows the situation when $h$ and $m$ are equal or below $0.5$. Despite having different accuracies, both plots show that the lines that define the condition to achieve complementary accuracy reside above the $\mathbb{E}[f^r] = \mathbb{E}[f^w]$ diagonal, and it is the differences of $h$ and $m$, rather than their absolute values, that determine the likelihood (the area above the line) of achieving complementary accuracy.

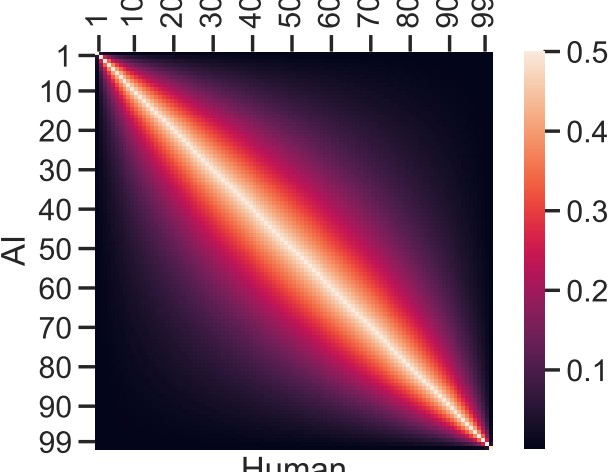

Figure 8: The heatmap showing the area above the $\mathbb{E}[f^w]$-$\mathbb{E}[f^r]$ line with respect to different values of human accuracy $h$ and AI accuracy $m$. The values of accuracy (in percentage) for human and AI are shown in the horizontal and vertical axes, and the color of the heatmap represents the area above the line of $\mathbb{E}[f^r]$ in Theorem 2. In Fig. 7, we illustrate different $\mathbb{E}[f^w]$-$\mathbb{E}[f^r]$ lines depending on different $h$ and $m$. The area above the line indicates the likelihood of achieving complementary performance for different combinations of $h$ and $m$. The heatmap shows that as the difference between $h$ and $m$ becomes smaller (near the diagonal), it permits more area above the line for achieving complementary accuaracy.

**Theorem 1-related results.** The previous experiment is in the condition where plausibility is correlated with AI prediction quality. What if plausibility is not correlated with AI prediction quality? We conduct the simulation experiment, which shows that while the rest conditions remain the same as in the previous experiment in Fig. 6, the generated plausibility values follow normal distributions and do not correlate with AI prediction correctness. The results are shown in Fig. 9. In either case when human accuracy is greater or less than AI accuracy, the complementary human-AI team accuracy cannot be achieved. This empirical finding corresponds to the theoretical finding of Theorem 1.

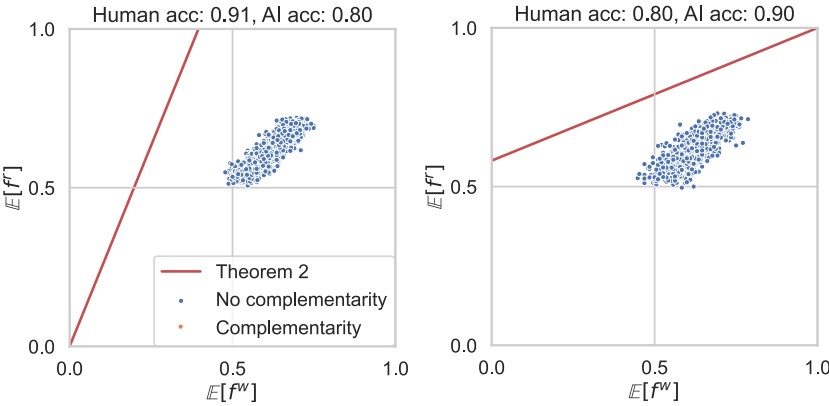

Figure 9: Visualization of the simulation experiment when plausibility does not correlate with AI prediction correctness. Each panel has 1000 dots, which represents 1000 simulation trials. Orange and blue dots indicate the complementary performance is achieved or not in a trial, respectively. The $\mathcal{L}$ line in Eq. (2) is shown in red to visualize the relationship between $\mathbb{E}[f^w]$ and $\mathbb{E}[f^r]$ in Theorem 2. The two plots show two conditions when human accuracy is greater or less than AI accuracy. In Fig. 6, we show similar plots. They only differ in that the simulation experiments in this figure draw the plausibility scores from normal distributions that are independent of AI prediction correctness, which follows the conclusion in Theorem 1 that complementary accuracy is not achieved.

# F Proof of the two theorems on plausibility and human-AI team performance

We provide proofs of the two theorems on plausibility and human-AI team performance in Section 3.3.2. We first set up the problem of human-AI collaboration, then provide proof for the two theorems regarding the influence of explanation plausibility to human-AI team performance.

We focus on the problem setting of human-AI collaboration where AI neither has full task delegation nor decides the task delegation, and acts as a decision assistant. This is a common scenario in human-AI collaboration especially in high-stakes tasks [51], and this is the scenario where AI explanation can play a major role. Otherwise, if AI decides the task delegation [60], there is little chance for AI explanation to play a role in either the task delegation or the whole decision-making process.

To simplify the problem, we use the task performance metric of accuracy for classification problems. It is worth noting that choosing different metrics may lead to different effects in explanation optimization, because different metrics emphasize different aspects that users think as important in performing a task, as shown in previous work [66]. We leave the exploration of using different task performance metrics on XAI optimization for future work.

**Problem setup**

The problem is in a collaborative setting where AI assists humans in making decisions on a task. For each case, the human decision-maker first reviews the AI suggestion, including the AI prediction and its explanation. Then the human decides whether to accept or reject AI assistance by judging how plausible the suggestion is based on human prior knowledge of the task. The more plausible an explanation is, the more likely the human will accept AI assistance and its suggestion. If AI assistance is rejected, the human delegates the decision-making task to herself and makes a final decision based on her own knowledge. Fig 10 illustrates this AI-assisted decision process.

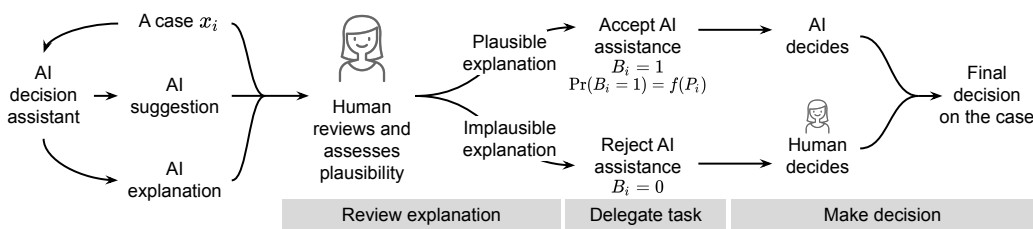

Figure 10: Flowchart of the AI-assisted decision-making workflow. The gray bars at the bottom highlight tasks that the human needs to perform.

We assume a test dataset $\mathcal{D} = \{x_1, \ldots, x_N\}$ has $N$ number of cases that are independent and identically distributed. We use the subscript $i \in [1, N]$ to denote the index of an instance in $\mathcal{D}$. For an instance $x_i \in \mathcal{D}$, we use $B_i \in \{0, 1\}$ to denote the binary random variable of human choosing to accept or reject the AI suggestion for $x_i$, with $B_i = 1$ representing "the human accepts the AI suggestion for $x_i$," and $B_i = 0$ representing "the human rejects the AI suggestion for $x_i$." $B_i$ follows a Bernoulli distribution with $\Pr(B_i = 1; f(P_i)) = f(P_i)$ and $\mathbb{E}[B_i] = f(P_i)$, where $\Pr$ denotes probability, $P_i \in \mathbb{R}$ denotes the random variable of the plausibility measure of an AI explanation $E_i$ for the prediction of $x_i$, and $f(P_i) \in [0, 1]$ denotes the probability of human acceptance of the AI suggestion for $x_i$. $f(.)$ is assumed to be a function of plausibility $P$ to denote the human factor function of the probability to decide to take an AI suggestion given the explanation plausibility $P$. In Theorem 2, we assume they have a causal relationship, $P \to f(P)$. An explanation with higher plausibility $P$ would lead to user's higher probability of acceptance $f(P)$. Based on the empirical data in Appendix E.1 on the causal correlation relationship between $P$ and $f(P)$, we have the following conjecture:

**Conjecture 1** (Relationship between Plausibility and Human Acceptance of AI). *For any instances $x_i, x_j$, the probability $f(P_i)$ of human acceptance of the AI suggestion for $x_i$ is a function of the explanation plausibility $P_i$ with a monotonically non-decreasing relationship: $\forall P_i = p_i, P_j = p_j$, if $p_i \geq p_j$, then $f(p_i) \geq f(p_j)$.*

We use $C_i \in \{0, 1\}$ to denote the binary random variable of an instance $x_i \in \mathcal{D}$ being predicted correctly or not by the decision-maker, with $C_i = 1$ representing "the instance $x_i$ is correctly predicted," and $C_i = 0$ representing "the instance $x_i$ is incorrectly predicted." $C_i$ follows a Bernoulli distribution, with $\Pr(C_i = 1; \gamma) = \gamma$ and $\mathbb{E}[C_i] = \gamma$, where $\gamma$ is the probability of $x_i$ being predicted correctly. When $x_i$ is predicted by the human, AI, or human-AI team, we use $C_i^{\text{human}}$, $C_i^{\text{AI}}$, and $C_i^{\text{team}}$ to denote the decision-maker of the random variable $C_i$, and denote $\gamma = h, m, t$, respectively. We use the random variable $K$ to denote the number of instances being correctly predicted in the dataset $\mathcal{D}$. Since $x_i \in \mathcal{D}$ are i.i.d., and $C_i$ denotes each $x_i$ being correctly predicted or not, $K$ follows a binomial distribution, with $\mathbb{E}[K] = \mathbb{E}[C_1 + \cdots + C_N] = \sum_{i=1}^{N} \mathbb{E}[C_i] = \sum_{i=1}^{N} \gamma = N\gamma$. Then the accuracy on the dataset $\mathcal{D}$ would be $\mathbb{E}[K]/N = \gamma$. The parameters $h, m$, and $t$ can also be used to denote the accuracies on the dataset $\mathcal{D}$ of the human, AI, or the human-AI team, respectively. We assume $h \in (0, 1)$ and $m \in (0, 1)$ to avoid the undefined case of division by zero.

**Definition 1** (Complementary Accuracy from Bansal et al. [8]). *In classification tasks, the complementary accuracy of a human-AI team is defined as the human-AI team accuracy $t$ being greater than either human accuracy $h$ or AI accuracy $m$ alone:*

$$t > \max(h, m).$$

Before we prove the two main theorems in the paper, we provide conditions to show when complementary accuracy is impossible to achieve. Negations of these conditions are the prerequisites for the following Theorems 1 and 2.

**Lemma 1** (Impossible Complementarity for Black-Box AI). *In classification tasks, let $h, m$, and $t$ be the accuracies of the human, AI, and human-AI team, respectively; the AI is a black-box AI that only provides the human with the predicted class label without any other information about the decision process for the data instance $x \in \mathcal{D}$, then the human-AI team can never achieve complementary accuracy, i.e.: $t \leq \max(h, m)$.*

*Proof.* For a data instance $x_i \in \mathcal{D}$, we use $b_i$ to denote the parameter of the Bernoulli distribution of the probability that "the human accepts the AI suggestion for $x_i$" with $\Pr(B_i = 1; b_i) = b_i$. Because the human is not provided with any information on the decision process, the random variable $B_i$ of human acceptance of AI suggestion is independent of the random variable of AI decision correctness $C_i^{\text{AI}}$. Then the joint probability of $\Pr(B_i, C_i^{\text{AI}})$ can be calculated by the multiplication of probabilities of individual events:

$$\Pr(B_i, C_i^{\text{AI}}) = \Pr(B_i)\Pr(C_i^{\text{AI}}).$$

Then for each instance $x_i$ in the test set, we can list the joint events of $B_i$ and $C_i^{\text{AI}}$, their joint probabilities, and their likelihood of being correctly predicted ($C_i = 1$) by the decision-maker in Table 5.

Table 5: Four events regarding the combinations of random variable assignments of $B_i$ and $C_i^{\text{AI}}$ for an instance $x_i$. $b_i$ is the probability of human accepting AI suggestion for instance $x_i$. Regarding the last column of the likelihood of the decision-maker correctly predicting $x_i$ ($C_i = 1$), for events i and ii, because the decision-maker is AI, and the events are conditioned on $C_i^{\text{AI}}$ being 1 or 0, therefore, $C_i = C_i^{\text{AI}}$. For events iii and iv, because the decision-maker is the human, then $C_i = C_i^{\text{human}}$.

| Event: $B_i$ and $C_i^{\text{AI}}$ | Values of the r.v. | Probability of the event: $\Pr(B_i, C_i^{\text{AI}})$ | Who is the decision-maker? | Likelihood of the decision-maker correctly predicting $x_i$ ($C_i = 1$) |
|---|---|---|---|---|
| i. Human accepts AI AI predicts correctly for $x_i$ | $B_i = 1$ $C_i^{\text{AI}} = 1$ | $b_i m$ | AI | 1 |
| ii. Human accepts AI AI predicts incorrectly for $x_i$ | $B_i = 1$ $C_i^{\text{AI}} = 0$ | $b_i(1-m)$ | AI | 0 |
| iii. Human rejects AI AI predicts correctly for $x_i$ | $B_i = 0$ $C_i^{\text{AI}} = 1$ | $(1-b_i)m$ | Human | $h$ |
| iv. Human rejects AI AI predicts incorrectly for $x_i$ | $B_i = 0$ $C_i^{\text{AI}} = 0$ | $(1-b_i)(1-m)$ | Human | $h$ |

We use $t_i$ to denote the probability of $C_i^{\text{team}} = 1$ for the human-AI team to correctly predict $x_i$. $t_i$ can be calculated by aggregating the likelihood of $C_i = 1$ from all the four potential events weighted by the corresponding probabilities of an event:

$$\begin{aligned}
t_i &= b_i m \times 1 + b_i(1-m) \times 0 + (1-b_i)mh + (1-b_i)(1-m)h \\
&= b_i m + (1-b_i)h \\
&= (m-h)b_i + h
\end{aligned}$$

The random variable $K$ denotes the number of instances being correctly predicted in the dataset $\mathcal{D}$. Then the human-AI team accuracy $t$ on the test set $\mathcal{D}$ can be calculated as:

$$t = \frac{\mathbb{E}[K^{\text{team}}]}{N} = \frac{\sum_{i=1}^{N} \mathbb{E}[C_i^{\text{team}}]}{N} = \frac{\sum_{i=1}^{N} t_i}{N}$$

$$= m\frac{\sum_{i=1}^{N} b_i}{N} + h(1 - \frac{\sum_{i=1}^{N} b_i}{N})$$

$$= (m - h)\frac{\sum_{i=1}^{N} b_i}{N} + h$$

If $h \geq m$:

$$t = (m - h)\frac{\sum_{i=1}^{N} b_i}{N} + h \leq (h - h)\frac{\sum_{i=1}^{N} b_i}{N} + h = h$$

$$\Rightarrow t \leq h$$

If $h < m$:

$$t = m\frac{\sum_{i=1}^{N} b_i}{N} + h(1 - \frac{\sum_{i=1}^{N} b_i}{N}) < m\frac{\sum_{i=1}^{N} b_i}{N} + m(1 - \frac{\sum_{i=1}^{N} b_i}{N}) = m$$

$$\Rightarrow t < m$$

Therefore, $t \leq \max(h, m)$

$\square$

In Lemma 1, the value of $b_i$ reflects the human interpretation of all available information provided by a black-box model, including the human interpretation of the input, the predicted label, and the overall performance of the AI model on a previous test set. Lemma 1 shows that with the limited and non-data-instance-specific information provided by black-box AI models, the black-box models are not equipped with the prerequisites to achieve complementary accuracy. Lemma 1 provides motivation for white-box and gray-box AI models that provide additional information about the model decision process, decision certainty, or decision quality. Such information can be the decision confidence or uncertainty estimation for the given instance (for example, the calibrated probability output for the predicted class), the fine-grained performance on different subsets of the data, an AI explanation, or a combination of these different types of information.

Other two conditions that make complementary performance impossible to achieve are identified in Donahue et al.'s work on theoretical investigation of complementarity and fairness [21], where they differed from our theoretical proofs by using the loss function as the performance metric, having a different set of assumptions on decision combination, and focusing on fairness rather than explainability. Lemma 3 in Donahue et al.'s work [21] states that "Complementarity is impossible if one of the human or algorithm always weakly dominates the loss of the other: that is, if $a_i \leq h_i$ for all $i$, or $a_i \geq h_i$ for all $i$," where $a_i$ and $h_i$ are the losses of AI and human for an instance $i$. We adapt the same conclusion to our problem setup and assumptions in Lemma 2.

**Lemma 2** (Adapted from Donahue et al. [21]). *If one decision-maker of either human or AI always dominates the prediction performance for all instances $x \in \mathcal{D}$, then the human-AI team can never achieve complementary performance.*

*Proof.* For any data instance $x_i \in \mathcal{D}$, since one decision-maker dominates the prediction performance for $x_i$, then the rational choice during task delegation is to delegate the decision-making task to the dominant decision-maker. Then the maximum task performance for the human-AI team is equivalent to the performance of the dominant decision-maker, i.e.: $\max t = \max(h, m)$. This concludes that complementary performance is impossible to achieve. $\square$

The last condition that makes complementary performance impossible is stated in Corollary 1 of Donahue et al.'s work [21]: "A combining function with a constant weighting function $w_h(a_i, h_i) = w_h$ can never achieve complementarity performance," where $w_h(.)$ is the weighting function of the human decision-maker "controlling how much the human influences the final prediction." The role of $w_h(.)$ is similar to $f(P)$ in our problem setup. Corollary 1 from [21] states that if the decision cobmination function has constant weights (i.e., the function $w_h(.)$ becomes a constant $w_h$) to

combine the human and AI decision-makers' loss for all instances, then it is impossible to achieve complementary performance. In our problem setup, we assume that the decision delegation (the probabilistic form is equivalent to the weighted decision combination in [21]) for each instance is an individual Bernoulli random process, with each instance $x_i$ having a different parameter of $f(P_i)$ of a different Bernoulli distribution. If $f(P_i)$ is the same for every instance $x_i$, i.e., $f(P_i) = \lambda$, then we can show in Lemma 3 that complementary performance is impossible to achieve.

**Lemma 3** (Adapted from Donahue et al. [21]). *If the AI suggestion has the same probability $\lambda$ to be accepted by human for every instance $x \in \mathcal{D}$, then the human-AI team can never achieve complementary performance.*

*Proof.* Since the probability of human acceptance of AI suggestion for any instance $x_i \in \mathcal{D}$ is a constant $\lambda$, then the human acceptance of AI suggestion is independent of the correctness of the decision-maker. Then we can use the same probability of the events in Table 5 by replacing $b_i$ in the table with $\lambda$. We use $t_i$ to denote the probability of $C_i^{\text{team}} = 1$ for the human-AI team to correctly predict $x_i$. $t_i$ can be calculated by aggregating the likelihood of $C_i = 1$ from all the four potential events weighted by the corresponding probabilities of an event.

$$
\begin{aligned}
t_i &= \Pr(C_i = 1) \\
&= \Pr(C_i = 1, B_i = 1) + \Pr(C_i = 1, B_i = 0) \\
&= \Pr(B_i = 1)\Pr(C_i = 1) + \Pr(B_i = 0)\Pr(C_i = 1) \\
&= \lambda m + (1 - \lambda)h
\end{aligned}
$$

The human-AI team accuracy $t$ on the test set $\mathcal{D}$ can be calculated as:

$$
\begin{aligned}
t &= \frac{\mathbb{E}[K^{\text{team}}]}{N} = \frac{\sum_{i=1}^{N} \mathbb{E}[C_i^{\text{team}}]}{N} = \frac{\sum_{i=1}^{N} t_i}{N} = t_i \\
&= \lambda m + (1 - \lambda)h
\end{aligned}
$$

$$
\begin{aligned}
&\text{If } h \geq m: \\
&\quad t = (m - h)\lambda + h \leq (h - h)\lambda + h = h \\
&\Rightarrow t \leq h \\
&\text{If } h < m: \\
&\quad t = m\lambda + (1 - \lambda)h < \lambda m + (1 - \lambda)m = m \\
&\Rightarrow t < m \\
&\text{Therefore, } t \leq \max(h, m)
\end{aligned}
$$

$\square$

In summary, from Lemma 1-3 we identify the conditions that are impossible to achieve complementary performance. Therefore, to potentially achieve complementary performance for the human-AI team, the AI models should be white-box or gray-box models that can provide additional information of the decision process to assist human judgment on whether to accept an AI suggestion, the human or AI should not always dominate the prediction performance, and the probability of human acceptance of AI suggestions should vary by data instances. These conditions set the prerequisites for the following Theorems 1 and 2 on when it is impossible or possible to achieve complementary accuracy with AI explanations.

Let us recall Theorem 1 from Section 3.3.2.

**Theorem 1** (Case of Impossible Complementarity for XAI). *Let $h$, $m$, and $t$ be the accuracies of the human, AI, and human-AI team, respectively; and $f(P_i)$ be a function of the explanation plausibility $P_i$ denoting the probability of human acceptance of the AI suggestion for the instance $x_i \in \mathcal{D}$, then:*

*If plausibility is independent of the AI decision correctness, then the human-AI team can never achieve complementary accuracy, i.e.: $t \leq \max(h, m)$.*

*Proof.* The procedure of proof is the same with the one for Lemma 1, with the only difference in that $b_i$ is replaced by $f(P_i)$.

If plausibility $P$ is independent of the AI decision correctness (denoted by the Bernoulli random variable $C^{\text{AI}}$) with $\Pr(P|C^{\text{AI}}) = \Pr(P)$, Because $\Pr(P_i|C_i^{\text{AI}}) = \Pr(P_i)$, and $f(P_i)$ is a function of $P_i$ with the specific function parameters determined by human factors that are independent of $C_i^{\text{AI}}$ (as humans have no access to the ground truth information that determines correctness), then $\Pr(f(P_i)|C_i^{\text{AI}}) = \Pr(f(P_i))$.

Because $f(P_i)$ is the only parameter that determines the Bernoulli distribution of $B_i$, then we can get $\Pr(B_i|C_i^{\text{AI}}) = \Pr(B_i)$. Then the joint probability of $\Pr(B_i, C_i^{\text{AI}})$ can be calculated by the multiplication of probabilities of individual events:

$$\Pr(B_i, C_i^{\text{AI}}) = \Pr(B_i|C_i^{\text{AI}})\Pr(C_i^{\text{AI}}) = \Pr(B_i)\Pr(C_i^{\text{AI}}).$$

Then for each instance $x_i$ in the test set, we can list the joint events of $B_i$ and $C_i^{\text{AI}}$, their joint probabilities, and their likelihood of being correctly predicted ($C_i = 1$) by the decision-maker in Table 6.

Table 6: Four events regarding the combinations of random variable assignments of $B_i$ and $C_i^{\text{AI}}$ for an instance $x_i$. $P_i$ is the plausibility of AI explanation for instance $x_i$, and $f(P_i)$ is the probability of human accepting AI suggestion for instance $x_i$ given $P_i$. Regarding the last column of the likelihood of the decision-maker correctly predicting $x_i$ ($C_i = 1$), for events i and ii, because the decision-maker is AI, and the events are conditioned on $C_i^{\text{AI}}$ being 1 or 0, therefore, $C_i = C_i^{\text{AI}}$. For events iii and iv, because the decision-maker is the human, then $C_i = C_i^{\text{human}}$.

| Event: $B_i$ and $C_i^{\text{AI}}$ | Values of the r.v. | Probability of the event: $\Pr(B_i, C_i^{\text{AI}})$ | Who is the decision-maker? | Likelihood of the decision-maker correctly predicting $x_i$ ($C_i = 1$) |
|---|---|---|---|---|
| i. Human accepts AI AI predicts correctly for $x_i$ | $B_i = 1$ $C_i^{\text{AI}} = 1$ | $f(P_i)m$ | AI | 1 |
| ii. Human accepts AI AI predicts incorrectly for $x_i$ | $B_i = 1$ $C_i^{\text{AI}} = 0$ | $f(P_i)(1-m)$ | AI | 0 |
| iii. Human rejects AI AI predicts correctly for $x_i$ | $B_i = 0$ $C_i^{\text{AI}} = 1$ | $(1 - f(P_i))m$ | Human | $h$ |
| iv. Human rejects AI AI predicts incorrectly for $x_i$ | $B_i = 0$ $C_i^{\text{AI}} = 0$ | $(1 - f(P_i))(1 - m)$ | Human | $h$ |

We use $t_i$ to denote the probability of $C_i^{\text{team}} = 1$ for the human-AI team to correctly predict $x_i$. $t_i$ can be calculated by aggregating the likelihood of $C_i = 1$ from all the four potential events weighted by the corresponding probabilities of an event:

$$\begin{aligned} t_i &= f(P_i)m \times 1 + f(P_i)(1-m) \times 0 + (1 - f(P_i))mh + (1 - f(P_i))(1-m)h \\ &= f(P_i)m + (1 - f(P_i))h \\ &= (m - h)f(P_i) + h \end{aligned}$$

The human-AI team accuracy $t$ on the test set $\mathcal{D}$ can be calculated as:

$$t = \frac{\mathbb{E}[K^{\text{team}}]}{N} = \frac{\sum_{i=1}^{N}\mathbb{E}[C_i^{\text{team}}]}{N} = \frac{\sum_{i=1}^{N} t_i}{N}$$

$$= m\frac{\sum_{i=1}^{N} f(P_i)}{N} + h(1 - \frac{\sum_{i=1}^{N} f(P_i)}{N})$$

$$= (m-h)\frac{\sum_{i=1}^{N} f(P_i)}{N} + h$$

If $h \geq m$:

$$t = (m-h)\frac{\sum_{i=1}^{N} f(P_i)}{N} + h \leq (h-h)\frac{\sum_{i=1}^{N} f(P_i)}{N} + h = h$$

$$\Rightarrow t \leq h$$

If $h < m$:

$$t = m\frac{\sum_{i=1}^{N} f(P_i)}{N} + h(1 - \frac{\sum_{i=1}^{N} f(P_i)}{N}) < m\frac{\sum_{i=1}^{N} f(P_i)}{N} + m(1 - \frac{\sum_{i=1}^{N} f(P_i)}{N}) = m$$

$$\Rightarrow t < m$$

Therefore, $t \leq \max(h, m)$

$\square$

Let us recall Theorem 2 from Section 3.3.2.

**Theorem 2** (Conditions for XAI Complementarity). *Let $h$, $m$, and $t$ be the accuracies of the human, AI, and human-AI team, respectively; $f(P_i)$ be the probability of human acceptance of an AI suggestion for an instance $x_i \in \mathcal{D}$, where $f(.)$ is a monotonically non-decreasing function of the explanation plausibility $P_i$; and $P_i^r$ and $P_i^w$ be the plausibility values of an AI explanation when an instance $x_i$ is predicted correctly or incorrectly, then:*

*Complementary human-AI accuracy can be achieved, i.e., $t > \max(h, m)$, when*

$$\begin{cases} h \geq m & and & \mathbb{E}[f^r] > \frac{h(1-m)}{m(1-h)}\mathbb{E}[f^w]; & or, \\ m > h & and & \mathbb{E}[f^r] > \frac{h(1-m)}{m(1-h)}\mathbb{E}[f^w] + \frac{m-h}{m(1-h)} \end{cases}$$

*where $\mathbb{E}[f^r]$ and $\mathbb{E}[f^w]$ are the expectations of $f(P_i^r)$ and $f(P_i^w)$ over the dataset $\mathcal{D}$, indicating among the correctly or incorrectly predicted instances of AI, how many are accepted by human.*

*Proof.* Since $P_i^r$ and $P_i^w$ are the plausibility values of an AI explanation when an instance $x_i \in \mathcal{D}$ is predicted correctly or incorrectly ($C_i^{\text{AI}} = 1$ or $0$), respectively, $P_i^r$ and $P_i^w$ are the shorthand notations for $P_i|C_i^{\text{AI}} = 1$ and $P_i|C_i^{\text{AI}} = 0$. Since $P_i$ is conditioned on $C_i$, and $f(P_i)$ is a function of $P_i$, then $f(P_i)$ is also conditioned on $C_i$. We use $f(P_i^r)$ to denote $f(P_i|C_i^{\text{AI}} = 1)$, and use $f(P_i^w)$ to denote $f(P_i|C_i^{\text{AI}} = 0)$.

Because $f(P_i)$ is the parameter that determines the Bernoulli distribution of $B_i$, and $f(P_i)$ is defined conditioned on $C_i$, then $B_i$ is also defined by conditioning on $C_i$, with:

$$\Pr(B_i = 1|C_i^{\text{AI}} = 1) = f(P_i|C_i^{\text{AI}} = 1) = f(P_i^r), \text{ and} \tag{3}$$

$$\Pr(B_i = 1|C_i^{\text{AI}} = 0) = f(P_i|C_i^{\text{AI}} = 0) = f(P_i^w) \tag{4}$$

With these, we can calculate the joint probability of $\Pr(B_i, C_i^{\text{AI}})$ by:

$$\Pr(B_i, C_i^{\text{AI}}) = \Pr(B_i|C_i^{\text{AI}})\Pr(C_i^{\text{AI}})$$

Then for each instance $x_i$ in the test set, we can list the joint events of $B_i$ and $C_i^{\text{AI}}$, their joint probabilities, and their likelihood of being correctly predicted ($C_i = 1$) by the decision-maker in Table 7.

Table 7: Four events regarding the combinations of random variable assignments of $B_i$ and $C_i^{\mathrm{AI}}$ for instance $x_i$. $P_i^r$ and $P_i^w$ are the plausibility of AI explanation for instance $x_i$ when AI predicts correctly or wrongly, and $f(P_i)$ is the probability of human accepting AI suggestion for instance $x_i$. Regarding the last column of the likelihood of the decision-maker correctly predicting $x_i$ ($C_i = 1$), for events v and vi, because the decision-maker is AI, and the events are conditioned on $C_i^{\mathrm{AI}}$ being 1 or 0, therefore, $C_i = C_i^{\mathrm{AI}}$. For events vii and viii, because the decision-maker is the human, then $C_i = C_i^{\mathrm{human}}$.

| Event: $B_i$ and $C_i^{\mathrm{AI}}$ | Values of the r.v. | Probability of the event: $\Pr(B_i, C_i^{\mathrm{AI}})$ | Who is the decision-maker? | Likelihood of the decision-maker correctly predicting $x_i$ ($C_i = 1$) |
|---|---|---|---|---|
| v. Human accepts AI AI predicts correctly for $x_i$ | $B_i = 1$ $C_i^{\mathrm{AI}} = 1$ | $f(P_i^r)m$ | AI | 1 |
| vi. Human accepts AI AI predicts incorrectly for $x_i$ | $B_i = 1$ $C_i^{\mathrm{AI}} = 0$ | $f(P_i^w)(1-m)$ | AI | 0 |
| vii. Human rejects AI AI predicts correctly for $x_i$ | $B_i = 0$ $C_i^{\mathrm{AI}} = 1$ | $(1 - f(P_i^r))m$ | Human | $h$ |
| viii. Human rejects AI AI predicts incorrectly for $x_i$ | $B_i = 0$ $C_i^{\mathrm{AI}} = 0$ | $(1 - f(P_i^w))(1-m)$ | Human | $h$ |

We use $t_i$ to denote the probability of $C_i^{\mathrm{team}} = 1$ for the human-AI team to correctly predict $x_i$. $t_i$ can be calculated by aggregating the likelihood of $C_i = 1$ from the four potential events weighted by the corresponding probabilities of the event:

$$
\begin{aligned}
t_i &= f(P_i^r)m + f(P_i^w)(1-m)0 + (1 - f(P_i^r))mh + (1 - f(P_i^w))(1-m)h \\
&= f(P_i^r)m + mh - f(P_i^r)mh + h - f(P_i^w)h - mh + f(P_i^w)mh \\
&= f(P_i^r)m - f(P_i^r)mh + h - f(P_i^w)h + f(P_i^w)mh
\end{aligned} \tag{5}
$$

The human-AI team accuracy $t$ on the test set $\mathcal{D}$ can be calculated as:

$$
\begin{aligned}
t &= \frac{\mathbb{E}[K^{\mathrm{team}}]}{N} = \frac{\sum_{i=1}^N \mathbb{E}[C_i^{\mathrm{team}}]}{N} = \frac{\sum_{i=1}^N t_i}{N} \\
&= m\frac{\sum_{i=1}^N f(P_i^r)}{N} - mh\frac{\sum_{i=1}^N f(P_i^r)}{N} + h - h\frac{\sum_{i=1}^N f(P_i^w)}{N} + mh\frac{\sum_{i=1}^N f(P_i^w)}{N}
\end{aligned} \tag{6}
$$

The terms $\frac{\sum_{i=1}^N f(P_i^r)}{N}$, $\frac{\sum_{i=1}^N f(P_i^w)}{N}$ are the expectations of $f(P_i^r)$ and $f(P_i^w)$:

$$
\frac{\sum_{i=1}^N f(P_i^r)}{N} = \mathbb{E}[f(P^r)] \tag{7}
$$

$$
\frac{\sum_{i=1}^N f(P_i^w)}{N} = \mathbb{E}[f(P^w)] \tag{8}
$$

We use $\mathbb{E}[f^r]$ and $\mathbb{E}[f^w]$ to simplify the notation. So Eq. (7) and Eq. (8) can be rewritten as:

$$
\frac{\sum_{i=1}^N f(P_i^r)}{N} = \mathbb{E}[f(P^r)] = \mathbb{E}[f^r] \tag{9}
$$

$$
\frac{\sum_{i=1}^N f(P_i^w)}{N} = \mathbb{E}[f(P^w)] = \mathbb{E}[f^w] \tag{10}
$$

Then Eq. (6) can be rewritten as:

$$t = m\mathbb{E}[f^r] - mh\mathbb{E}[f^r] + h - h\mathbb{E}[f^w] + mh\mathbb{E}[f^w] \tag{11}$$

The meaning of $\mathbb{E}[f^r]$ and $\mathbb{E}[f^w]$ can be interpreted as follows:

If we use the definition of $f(P_i^r)$ and $f(P_i^w)$ in Eq. (3) and Eq. (4), then the term $\mathbb{E}[f^r]$ and $\mathbb{E}[f^w]$ can be written as:

$$
\begin{aligned}
\mathbb{E}[f^r] &= \frac{\sum_{i=1}^{N} f(P_i^r)}{N} \\
&= \frac{\sum_{i=1}^{N} f(P_i | C_i^{\mathrm{AI}} = 1)}{N} \\
&= \frac{\sum_{i=1}^{N} \mathsf{Pr}(B_i = 1 | C_i^{\mathrm{AI}} = 1)}{N}
\end{aligned} \tag{12}
$$

$$
\begin{aligned}
\mathbb{E}[f^w] &= \frac{\sum_{i=1}^{N} f(P_i^w)}{N} \\
&= \frac{\sum_{i=1}^{N} f(P_i | C_i^{\mathrm{AI}} = 0)}{N} \\
&= \frac{\sum_{i=1}^{N} \mathsf{Pr}(B_i = 1 | C_i^{\mathrm{AI}} = 0)}{N}
\end{aligned} \tag{13}
$$

$\mathbb{E}[f^r]$ means, among the correctly predicted instances ($C_i^{\mathrm{AI}} = 1$), how many are accepted by human ($B_i = 1$); Similarly, $\mathbb{E}[f^w]$ means, among the incorrectly predicted instances ($C_i^{\mathrm{AI}} = 0$), how many are accepted by human ($B_i = 1$). In this sense, $\mathbb{E}[f^r]$ is a measure of sensitivity (true positive rate), and $\mathbb{E}[f^w]$ is a measure of false positive rate.

From Eq. (11), we can get the conditions for complementary accuracy as follows:

If $h \geq m$:
$$
\begin{aligned}
t - h &= m\mathbb{E}[f^r] - mh\mathbb{E}[f^r] + h - h\mathbb{E}[f^w] + mh\mathbb{E}[f^w] - h \\
&= m\mathbb{E}[f^r] - mh\mathbb{E}[f^r] - h\mathbb{E}[f^w] + mh\mathbb{E}[f^w] \\
&= m(1-h)\mathbb{E}[f^r] - h(1-m)\mathbb{E}[f^w]
\end{aligned}
$$

If $\mathbb{E}[f^r] > \dfrac{h(1-m)}{m(1-h)}\mathbb{E}[f^w]$,

then $m(1-h)\mathbb{E}[f^r] - h(1-m\mathbb{E}[f^w]) > 0$

then $t - h > 0$ given $h \geq m$ and $\mathbb{E}[f^r] > \dfrac{h(1-m)}{m(1-h)}\mathbb{E}[f^w]$

If $m > h$:
$$
\begin{aligned}
t - m &= m\mathbb{E}[f^r] - mh\mathbb{E}[f^r] + h - h\mathbb{E}[f^w] + mh\mathbb{E}[f^w] - m \\
&= m(1-h)\mathbb{E}[f^r] - h(1-m)\mathbb{E}[f^w] - (m-h)
\end{aligned}
$$

If $\mathbb{E}[f^r] > \dfrac{h(1-m)}{m(1-h)}\mathbb{E}[f^w] + \dfrac{m-h}{m(1-h)}$,

then $m(1-h)\mathbb{E}[f^r] - h(1-m)\mathbb{E}[f^w] - (m-h) > 0$

then $t - m > 0$ given $m > h$ and $\mathbb{E}[f^r] > \dfrac{h(1-m)}{m(1-h)}\mathbb{E}[f^w] + \dfrac{m-h}{m(1-h)}$

Therefore,

$$
t > \max(h, m) \text{ if }
\begin{cases}
h \geq m \text{ and } \mathbb{E}[f^r] > \dfrac{h(1-m)}{m(1-h)}\mathbb{E}[f^w] \text{ , or} \\
m > h \text{ and } \mathbb{E}[f^r] > \dfrac{h(1-m)}{m(1-h)}\mathbb{E}[f^w] + \dfrac{m-h}{m(1-h)}
\end{cases}
\tag{14}
$$

$\square$

From Theorem 2, we can get the following corollaries.

**Corollary 1.** *If a human-AI team can achieve complementary accuracy, then the human acceptance rate for correctly predicted data should be greater than the human acceptance rate for incorrectly predicted data, $\mathbb{E}[f^r] > \mathbb{E}[f^w]$. Furthermore, the mean plausibility for correctly predicted data should be greater than the mean plausibility for incorrectly predicted data, $\mathbb{E}[P^r] > \mathbb{E}[P^w]$.*

*Proof.* To achieve complementary human-AI accuracy, it should fulfill one of the two conditions in Eq. (14).

When $h \geq m$,

$$h - hm \geq m - hm$$

$$\frac{h(1-m)}{m(1-h)} \geq 1$$

Therefore,

$$\mathbb{E}[f^r] > \frac{h(1-m)}{m(1-h)}\mathbb{E}[f^w] \geq \mathbb{E}[f^w]$$

When $m > h$,

$$\frac{h(1-m)}{m(1-h)}\mathbb{E}[f^w] + \frac{m-h}{m(1-h)} - \mathbb{E}[f^w] = (\frac{h(1-m)}{m(1-h)} - 1)\mathbb{E}[f^w] + \frac{m-h}{m(1-h)}$$

$$= \frac{h-m}{m(1-h)}\mathbb{E}[f^w] + \frac{m-h}{m(1-h)}$$

$$= \frac{m-h}{m(1-h)}(1 - \mathbb{E}[f^w]) \geq 0$$

Therefore,

$$\mathbb{E}[f^r] > \frac{h(1-m)}{m(1-h)}\mathbb{E}[f^w] + \frac{m-h}{m(1-h)} \geq \mathbb{E}[f^w]$$

And according to Conjecture 1, because $P$ and $f(P)$ have the monotonically non-decreasing relationship, then

$$\mathbb{E}[P^r] > \mathbb{E}[P^w]$$

$\square$

Corollary 1 indicates that to achieve complementary human-AI performance, the difference between $\mathbb{E}[f^r]$ and $\mathbb{E}[f^w]$, and accordingly, the plausibility for correct and incorrect decisions $P_i^r$ and $P_i^w$, should be big enough, i.e., above a threshold. Such relationships of $\mathbb{E}[f^r] > \mathbb{E}[f^w]$ and $\mathbb{E}[P^r] > \mathbb{E}[P^w]$ are necessary but not sufficient conditions to achieve complementary human-AI performance.

**Corollary 2.** *If complementary human-AI accuracy is achievable for an AI model, then with the assistance of this AI model, both novices and experts can achieve complementary accuracy despite their differences in prior knowledge.*

*Proof.* Eq. 14 does not impose constraints on the level of human performance $h$, therefore, complementary human-AI accuracy is achievable for both novices and experts as long as they have the domain knowledge for the given task that allows them to provide a reasonable estimate of $P$ [17]. $\square$

Corollary 2 also indicates that since $f(P)$ is dependent on the human judgment of $P$, novices and experts may have different net increases in complementary human-AI accuracy $t - \max(h, m)$ from the AI system, due to their different assessments of $P$ and $f(P)$ accordingly.

**Corollary 3.** *It is possible for both an inferior and a superior AI to help humans achieve complementary human-AI accuracy.*

*Proof.* Theorem 2 shows both conditions for AI that is either superior ($m > h$) or inferior ($h \geq m$) to human in accuracy to help human achieve complementary human-AI accuracy. $\square$

Corollary 3 indicates that as long as the explanation plausibility can be highly indicative of the AI decision correctness, even with the assistance of an inferior AI, humans can still benefit from the inferior AI and achieve complementary human-AI performance. However, as the machine accuracy $m$ decreases, the ratio of $\frac{\mathbb{E}[f^r]}{\mathbb{E}[f^w]} = \frac{h(1-m)}{m(1-h)}$ increases, which indicates that the difference — between the plausibility of correctly and incorrectly predicted instances — should be bigger to achieve complementarity.

**Limitation analysis**

A limitation in our analysis of the complementary human-AI task performance in this section is that, to model AI-assisted decision-making and task performance with AI explanations, in our problem setup in Fig. 10, we only utilize the simplest setup where the user delegates the task to AI or herself as a binary decision. And if the user delegates the task to herself, her decision-making is independent of the AI suggestion. In reality, unless otherwise instructed, a user may not accept or reject an AI suggestion in a binary fashion, and may include or exclude AI's second opinion as a decision option[9] in a probabilistic manner depending on plausibility and other factors of AI trustworthiness. Future works can explore various task delegation settings for XAI in AI-assisted decision making, and whether and how the ways of collaboration will influence complementary human-AI task performance.

# G    Analysis of examples on plausibility assessment and misleading explanations

We provide a detailed analysis of the examples in Fig. 1 of the paper, to show the subtle differences in human- and computationally-assessed plausibility and the role of human prior knowledge. In Fig. 1 of the paper, we give four examples that cover different combinations of plausible/implausible reasons for correct/incorrect predictions. The examples are on a task to classify bees vs. flies. We use an input image with the ground truth label of an Osmia ribifloris bee (Fig. 12). The AI explanations are given in the form of important feature set A, where the important features are expressed by a feature localization mask on the input image and a text description. This explanation form can be generated using a combination of the forms of a saliency map explanation [41] and concept explanation [45].

## G.1    The analytical framework: explanation is an explanatory argument with three propositions

Since plausibility is related to the human interpretation of explanation, we first detail the analytical framework we introduced in Section 3.3.3 on how humans make sense of a conclusion given an explanation. We regard an explanation as an argument that provides reasons for this question: *why is the input X predicted as the output Y?* And humans' interpretation of a given explanation is in a deductive manner. We apply syllogism in logical reasoning to analyze the human interpretation of explanation. For different explanation forms in predictive tasks, including saliency map, concept, prototype, example, and rule-based explanations, they have a common element of presenting the evidence of prediction in the form of features. In a syllogistic view, the feature set A is the middle term, input $X$ is the minor term, and output $Y$ is the major term. Then, a general form of explanatory argument is the following:

| | | |
|---|---|---|
| Proposition ① | $X$ has A. | Minor premise |
| Proposition ② | A is the set of important features for $Y$. | Middle term |
| Proposition ③ | A is discriminative for Y. | Major premise |
| | $X$ is predicted to be $Y$. | Conclusion |

The above form slightly differs from the standard form of a syllogism, as we separate the feature set A from the major premise (proposition ③) as a standalone proposition ② that states: A is the set of important features for the prediction $Y$. And ③ further states the detailed inference process on how A is discriminative for $Y$. Making A a standalone proposition is to facilitate the assessment of plausibility.

---

[9]For example, in doctor's differential diagnosis [87].

This form dissects human's interpretation process of an explanation so that we can analyze each proposition for plausibility. Plausibility denotes a person's judgment of the degree of an argument or proposition being true according to the person's knowledge [10]. The human assessment of plausibility thus includes the plausibility judgment of all three propositions being true. And the computational assessment of plausibility includes the plausibility judgment of proposition ② being true.

In AI explanation, the main information is the feature set A, and the two premises are not always given by AI explanation. According to the ostensive-inferential model in human communication, premises are context, which is the audience's assumption of the world [76]. When contextual information is lacking, users have to use their knowledge to infer the most probable premises given the evidence presented in the features. Therefore, it depends on the audience's assumptions and knowledge to infer the premises and their level of plausibility. Since human inference relies on human prior knowledge, the audience's inferential process may not be faithful to the model's underlying inference process, unless an explicit machine inferential process is provided by the AI explanation.

## G.2 Four examples presenting different combinations of the degree of plausibility and decision correctness

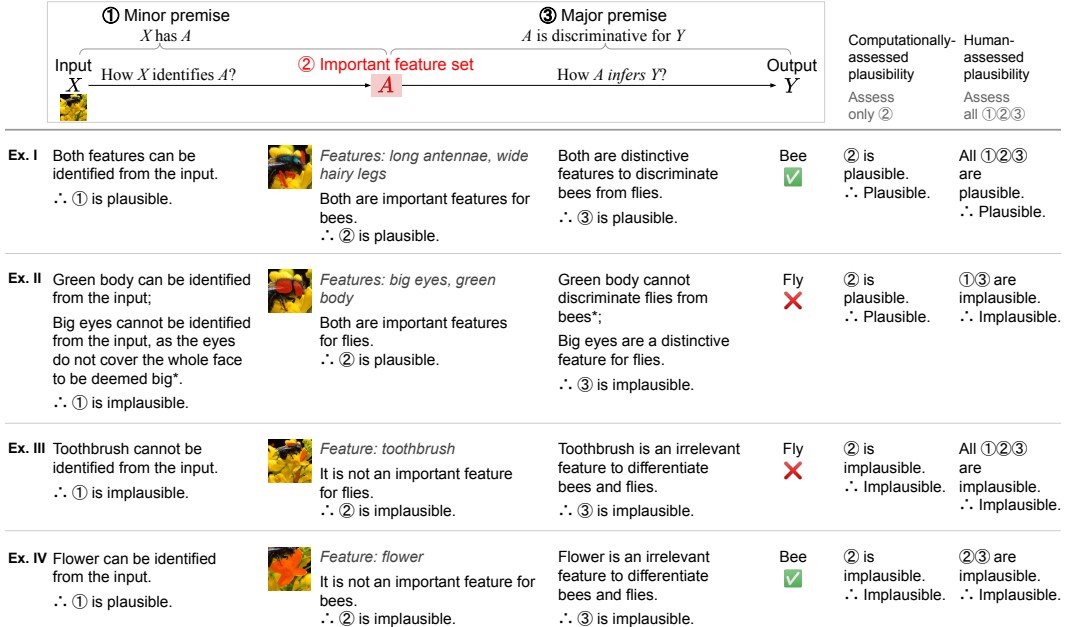

Figure 11: Analysis of the four examples in Fig. 1 of the paper regarding computationally- and human-assessed plausibility.

We provide an analysis of the four examples based on the above framework, which separates the plausibility of an explanation into the plausibility of the three propositions, illustrated in the top row of Fig. 11. Example (Ex.) I is a plausible explanation for a right prediction; Ex. II is a plausible explanation for a wrong prediction; Ex. III is an implausible explanation for a wrong prediction; And Ex. IV is an implausible explanation for a right prediction. Here, the plausible or implausible explanations are assessed computationally on the feature set A only.

For computationally-assessed plausibility, it calculates the similarity between humans' and AI's important feature set A to the prediction $Y$, which is the plausibility of proposition ②. In Ex. I and II, A is plausible because it identifies the characteristic body parts of the insect. In Ex. III and IV, A is not plausible because it focuses on the background rather than the insect.

___

[10]Strictly speaking, the truth and falsehood judgment can only apply to a proposition, not an argument. And the judgment of the faithfulness of an argument is termed soundness [36]. In the assessment of plausibility, we do not emphasize the distinction between a proposition and an argument.

For human-assessed plausibility, in addition to assessing the plausibility of proposition ②, a human will also assess propositions ① and ③. Such information is not provided by AI explanations in our examples, and is mainly inferred by the users. Proposition ① states *"how the feature set* A *can be identified from the input* $X$*."* Features in Ex. I (long antennae, wide hairy legs) and IV (flower) are plausible because they can be directly localized from the input image. Ex. II has two features: a green body and big eyes. Although the saliency map correctly localizes both features, the feature of big eyes cannot be identified from $X$, as the eyes are not big enough to cover the whole face, which is a criterion that distinguishes flies from bees. Note that such information requires some in-depth domain knowledge, which we mark with a ∗. Whether one possesses such knowledge or not makes a difference in the assessment of plausibility. For Ex. III, although the saliency map highlights the location of the feature, it cannot be recognized as a toothbrush. Therefore, the toothbrush feature cannot be identified from $X$, and proposition ① is implausible.

Proposition ③ states *"how the feature set* A *is discriminative features for the prediction* $Y$*."* Human knowledge is used to both infer the most possible premise that constructs the proposition based on the provided A, and judge the plausibility of the proposition. A in Ex. I provides distinctive features (long antennae and wide hairy legs) to discriminate bees from flies, thus this proposition is plausible. In Ex. II, the feature of a green body is not a distinguishing feature for flies, and can be a characteristic of some bees as well; the feature of big eyes that cover the entire face is a distinguishing feature for flies. Because the green body feature is implausible, the whole proposition is implausible. In Ex. III and IV, both features (toothbrush and flowers) are irrelevant features to differentiate bees and flies, thus both propositions are implausible.

With the above analysis of the plausibility of each proposition in the four examples, we have plausibility of an explanation assessed by human or machine, as shown in Fig. 11. There is a discrepancy between the two ways of assessment in Ex. II: the explanation is plausible by only assessing the feature set A; but when humans carefully examine its premises ① and ③, we will identify flaws in its argument that deem it implausible. A person without in-depth domain knowledge could also judge premises ① and ③ as plausible. This is a misleading explanation that misleads users to take the wrong suggestions of AI with its seemingly plausible explanation. We discuss misleading explanations in the next two subsections.

### G.3 Where do misleading explanations come from?

From the above analysis of the four examples, we can see misleading explanations (plausible explanations for wrong predictions) exist because the computational assessment of plausibility cannot well distinguish plausible explanations from implausible ones. The computational assessment of plausibility can only assess the plausibility of feature set A, but not the contextual information of the premises (propositions ① and ③) that are inferred by human audiences. Only human-assessed plausibility may sometimes be able to identify the unreasonableness of misleading explanations.

Even with human-assessed plausibility, misleading explanations may still be unavoidable due to humans' or AI's epistemic gaps: 1) As shown in Ex. II, users may lack in-depth domain knowledge to discern misleading explanations; 2) The AI model may not know it is predicting incorrectly despite the best effort to calibrate its decision certainty. Even though misleading explanations may not be eliminated, we cannot increase the number of misleading explanations to exacerbate this issue. As we have argued in the paper, using plausibility to evaluate or optimize XAI algorithms will increase the percentage of misleading explanations, which should be avoided.

### G.4 What are the dangers of misleading explanations?

In some tasks, misleading explanations may not be a big concern if humans can clearly recognize the misleading explanation being implausible by incorporating contextual information from human prior knowledge, as we show in the analysis of Ex. II in Fig. 11. This typically happens when the task is not ambiguous, very easy for humans to perform, or humans have complete information or knowledge about the task. However, such ideal scenarios are not always the case in real-world tasks, especially in cases where AI explanations are needed.

First, the common triggering motivations for users to check AI explanations include: resolving disagreements between users and AI, verifying AI suggestions to ensure the safety and reliability of decisions, detecting biases, improving user's own skills and knowledge, or making new discov-

eries [39]. For scenarios where users need AI explanations the most, they usually do not meet the above conditions that allow users to easily recognize misleading explanations.

Second, identifying misleading explanations requires in-depth domain knowledge (such as the knowledge of how big the eyes should be for a fly in Fig. 11 Ex. II) with the complete information provided for a task (such as the right perspective of the photo to capture the characteristics of the insect), as we show in the analysis of Ex. II. There are many real-world tasks where humans or AI would not have access to complete information, and need to make decisions under limited information, such as medical or financial decisions. In this scenario, it may be difficult for users to discern misleading explanations given incomplete information of the task or users' lack of in-depth domain knowledge.

Third, even if users can potentially discern misleading explanations, misleading explanations can still make the evidence for incorrect decisions more accessible to users than the evidence for correct decisions. It may cause users to overweigh and latch onto the evidence for wrong decisions. This is the anchoring effect in human judgment [82, 74].

Therefore, the dangers of misleading explanations are that they have negative impacts on users' decision correctness and task performance as stated in the paper, and may not be easily recognizable in real-world tasks.

The fallacy of misleading explanations is that they use seemingly plausible explanations to support the wrong decisions. In logic, this is an invalid argument as it breaks the logical link between true premises and true conclusion. In this sense, the plausibility of explanation acts as an indicator for decision certainty or confidence. And we should set the same goal for plausibility of XAI algorithms as uncertainty estimation [1] or confidence calibration [29], to avoid the model confidently being wrong.

# H Additional figure

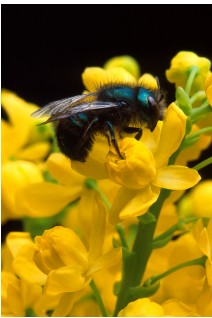

Figure 12: The original image used in Fig. 1, Fig. 2, and Fig. 11 of the paper. Photo of an Osmia ribifloris bee on a barberry flower. Photo by Jack Dykinga, USDA Agricultural Research Service. Public domain image, image source link: https://www.ars.usda.gov/oc/images/photos/may00/k5400-1/.

