# OpenReview forum: "Position: Why is plausibility surprisingly problematic as an XAI criterion?"
_NeurIPS.cc/2025/Position_Paper_Track — Submitted to NeurIPS 2025 Position Paper Track_

### Official Review · Reviewer_ybgB · 2025-07-20

**Significance:** 2
**Presentation:** 2
**Rating:** 1
**Confidence:** 5

**Summary:**

This position paper argues that plausibility (i.e., how convincing AI explanations appear to humans) should not be used as a primary criterion for evaluating explainable AI (XAI) algorithms. The authors demonstrate that optimizing for plausibility increases "misleading explanations" (convincing explanations for incorrect predictions), which manipulate users, erode trust, and prevent effective human-AI collaboration. Instead, they propose using plausibility as an intermediate measure to assess XAI's specific intended purposes rather than as an end goal, emphasizing that human explanations should not serve as ground truth for XAI evaluation.

**Strengths:**

I apologise in advance but this paper has modified the latex template and hence I feel it should not progress to the reviewing stage. To treat it otherwise would be unfair to the other submissions.

The section headings (e.g., the Introduction) have not enough white space above them, this happens a lot throughout the paper.

Anyone who compares Page 1 to any other submission will see what I mean, it is quite obvious.


### Nevertheless, here are the paper's strenghts in my view:
The paper argues addresses an important topic, and argue against the use of plausibility as a primary evaluation criterion. The position is articulated clearly, supported by theoretical analysis, illustrative examples, and references to prior literature. The authors also propose constructive alternatives, such as using plausibility as an intermediate measure tied to specific user-oriented purposes, and provide mathematical conditions for complementary human–AI performance, which could guide future research.

**Weaknesses:**

The core argument that plausible explanations are not necessarily correct may be viewed as intuitive, and not that interesting of a position.

Plausibility is after all badly defined.

**Questions:**

How do you explain the formatting issues?

**Alternative Position:**

Yes, and alternative positions are well-considered and addressed by the argument

**Author Identification:**

No.

**Context:**

2

**Discussion:**

2

**Ethics:**

["NO or VERY MINOR ethics concerns only"]

**Position:**

Yes, the paper argues for or against a position related to machine learning.

**Support:**

2

**Thoroughness:**

2

---

### Official Review · Reviewer_qA5o · 2025-08-08

**Significance:** 3
**Presentation:** 3
**Rating:** 7
**Confidence:** 4

**Summary:**

The paper considers plausibility for XAI, and argues strongly against and suggests elimination entirely.  This includes discussion, referencing examples, theory, and experiments.

**Strengths:**

Provides specific position(s) on use of plausibility for XAI, and argues forcefully in various ways for this, considering examples from literature, theory, and experiment.

Paper provides in depth discussions and is well referenced.

Relates plausibility to other XAI criteria such as transparency.

Extensive appendix includes Theorem proofs, experiments, and more.

Figure 1 encapsulates much of the papers arguments in a well conceived and clear way.

**Weaknesses:**

Seems to boil down to the fact that a plausible explanation is not necessarily a correct one.  To the reviewer this seems pretty obvious (although the paper presents a lot of evidence that this is apparently not true across the ML community!).

For a position paper, perhaps much of the appendix is better in a 'technical' paper track (??).

Section 3.3.1:  It seems there may be some tradeoffs in XAI performance and overall AI performance, e.g., AI architecture.

Theorem 1: should say in body of paper what is the model and key assumptions and conjecture 1 (which is a strong questionable assumption).

Conclusion: call on the community to stop using plausibility as **the** XAI criterion. Nevertheless, with sufficiently defined metrics, and sufficiently educated users, it might be used as one of several ways to understand an AI output.  But clearly plausibility alone is insufficient.

Paper doesn’t consider advanced uses of AI as an assistant, and focuses on one-time binary decision problems.  However, AI is becoming an interactive tool, able to provide multiple possible solutions, enable iteration with a user including querying, provide for ‘insufficient evidence’ type conclusions, and provide confidence and error metrics.

**Questions:**

Why is the title a question and not a position? (Yes, the reviewer is answering a question with a question in a moment of lightheartedness.)

Is eqtn (1) the only way to define plausibility in the AI context?

Perhaps instead we should ask 'why is that plausible' in an interactive-adaptive context?

Is much of this down to different definitions of plausible, and especially different engineering-quantifiable measures versus semantic-verbal definitions?

Theorem 1, conjecture.  How does a human only judge plausibility without also judging probability of correctness (or at least biased by this)?

**Alternative Position:**

Yes, and alternative positions are trivial straw-man arguments

**Author Identification:**

No.

**Context:**

3

**Discussion:**

4

**Ethics:**

["NO or VERY MINOR ethics concerns only"]

**Position:**

Yes, the paper argues for or against a position related to machine learning.

**Support:**

3

**Thoroughness:**

4

---

### Official Review · Reviewer_YLNi · 2025-08-14

**Significance:** 4
**Presentation:** 4
**Rating:** 7
**Confidence:** 4

**Summary:**

The paper argues that plausibility (defined by how similar an AI explanation is to a human explanation) should not be used to optimize and evaluate XAI algorithms. The main reasons are: 1) Human explanations are not the ground truth for XAI, so plausibility is invalid to measure explanability. 2) Using plausibility as an XAI criterion destroys trust, manipulates users, and cannot achieve complementary human-AI performance. 3) Plausible features are a sufficient but not necessary condition of understandability, so focusing on plausibility ignores other possibilities of enhancing understandability. The authors then suggest that instead of using plausibility as an end, researchers should use it as a means for other purposes, such as decision verification and bias detection.

**Strengths:**

The paper addresses an important topic that is very relevant to the XAI community. Evaluation criteria guide the future direction of the XAI field, so this topic is worthy of discussion at the conference.

The paper's arguments are well-supported by proofs and evidence, including the comprehensive appendices. The opposing opinions are thoroughly addressed and refuted. The presentation is clear. The examples and figures help illustrate the idea, and the authors provide the necessary definitions and background information, making the paper accessible to a wider audience.

**Weaknesses:**

In the paragraph starting on line 85, Reason 5, 7, and 8 of the alternative view do not have citations. I am wondering whether there are papers that advocate for those reasons.

**Questions:**

In the abstract, the authors explain plausibility as "how convincing the AI explanation is to humans." Then, the formal definition given in section 2 is $P = \text{similarity}(E^\text{human}, E^\text{AI})$. I am wondering whether the similarity between an AI explanation and a human explanation is the only factor affecting how convincing the AI explanation is to humans.

**Alternative Position:**

Yes, and alternative positions are well-considered and addressed by the argument

**Author Identification:**

No.

**Context:**

3

**Discussion:**

3

**Ethics:**

["NO or VERY MINOR ethics concerns only"]

**Position:**

Yes, the paper argues for or against a position related to machine learning.

**Support:**

4

**Thoroughness:**

4

---

### Note · Authors · 2025-08-27

**1-11 Submit Again:**

Definitely yes

**1-1 Submission Process:**

4

**1-2 Next Year:**

Critiques and critical examinations of existing practice, standards, and norms are essential components to promote the healthy development of a research community. We look forward to seeing the position paper track venue become a critical venue to hold such critical debates and dialogues on AI research and development. We’re especially interested in the idea of "Structured debates on controversial topics". Because this can showcase different perspectives, values, and opinions to simulate thoughts. In addition to the in-person debate during the conference, and offline discussions on OpenReview, we would also like to suggest an idea to promote more formal discussion on controversial topics. Open Peer Commentaries could be such a format to encourage wide discussion in a more formal format. People from within and outside the AI technical community can write a commentary for a target position paper, that will be published together with the position paper. The position paper track committee can refer to details on it: “Open Peer Commentaries: A Guide to the Perplexed” https://www.tandfonline.com/doi/full/10.1080/21507740.2025.2519452

**1-3 Future Development:**

We encountered a formatting issue with the latex template and have described the issue in our response #2 to Reviewer ybgB. In the NeurIPS latex template, the automatic adjustment did not allow breaking the paragraph that follows a new (sub)section title into two parts to be displayed on two consecutive pages. Instead, it forcefully puts the whole new (sub)section on a new page. This left excessive blank spaces on the previous page, either at the bottom or in-between paragraphs.

Since the paper has page limit, not word limit, we thought carefully about the problem we encountered and would like to kindly suggest the following ideas to improve the paper submission experience for future authors for the position paper track:

1. The conference submission can provide a Word template along with the Latex template. As far as we know, the Word editor can automatically break the paragraph to fill the excessive blank space, and won’t forcefully move the new section to the next page. Including a Word template may also encourage the inclusiveness of conference submission, especially for the position paper track. It can encourage dialogue and engagements from other disciplines and from civil society (especially from marginalized communities and the global south) beyond the AI community alone, who may not have the technical skill of latex writing or have access to latex.

2. The conference committee can provide a document for latex best practice along with the existing Instruction, to highlight what are the do's and don'ts when formatting latex. For example, here’s [the Latex best practice from ACM submission](https://www.acm.org/binaries/content/assets/publications/taps/latex-best_practices-06-may-2020.pdf).

**1-4 Interest:**

["Panel discussions with other position paper authors", "Structured debates on controversial topics", "Workshops for developing position papers", "Mentorship programs for early-career researchers"]

**1-5 Thoughtful:**

9

**1-6 Supportive:**

10

**1-7 Technical Aspects Versus Position:**

7

**1-8 Gate Keeping:**

8

**1-9 Camera Ready Changes:**

1. Add the following references for Reasons 5,7,8, according to our response #1 to Reviewer YLNi.

Reason 5 Plausibility and transparency
Ref: “Bringing Transparency Design into Practice” https://dl.acm.org/doi/10.1145/3172944.3172961. In Design Evaluation, it uses plausibility for the evaluation of the transparency design.

Reason 7 Plausibility and AI-assisted task performance
Ref: “A Human-Grounded Evaluation of SHAP for Alert Processing” https://arxiv.org/pdf/1907.03324. In Hypothesis 1, it hypothesizes that plausibility can be helpful for task accuracy.

Reason 8. Plausibility and understandability
Ref: “Opportunities and Challenges in Explainable Artificial Intelligence (XAI): A Survey” https://arxiv.org/abs/2006.11371. Fig. 19 contains a case study that uses plausibility to compare how understandable different XAI algorithms are.

2. Add a footnote in Sec. 2 to clarify the difference between human and computational assessment/definition of plausibility. For details, please see our response #2 to Reviewer YLNi.

3. In Theorems 1 & 2, we will briefly describe the task, key assumptions, and Conjecture 1 in the main text, according to Reviewer qA5o’s suggestion. These contents are originally presented in the Appendix.

4. We will add a sentence in Conclusion describing that, regarding the recent developments of advanced AI especially LLMs, as long as the explanation for LLMs falls within the scope and definition of plausibility, our analysis and conclusions apply to them as well. Thanks for Reviewer qA5o’s reminder.

5. Remove \vspace in latex, according to our response #2 to Reviewer ybgB.

**3-1 Review Response1:**

YLNi

**3-2 Reaction To Review1:**

1. On Weaknesses, we thank the reviewer for pointing out the missing references for Reason 5, 7, and 8. The following references will be added:

Reason 5: Eiband 2018. Bringing Transparency Design into Practice
Reason 7: Weerts 2019. A Human-Grounded Evaluation of SHAP for Alert Processing
Reason 8: Das 2020. Opportunities and Challenges in Explainable Artificial Intelligence (XAI): A Survey

2. On Questions, no, the similarity between human and AI explanation isn't the only factor affecting how convincing the AI explanation is according to human prior knowledge. The other factor that affects explanation convincingness is human tacit knowledge. Tacit knowledge is a type of human prior knowledge that may not necessarily be expressible in explanatory format such as language, thus it may not be included in the more formal expression format of human explanation. The discrepancy between the two definitions of plausibility is because explanation convincingness or reasonableness is a thick concept that describes and evaluates simultaneously, and Eq 1 gives its operationalized definition. We discussed and illustrated the discrepancy in Example 2, Fig 11 in Appendix G on pg 37, where the original thick definition assesses the three propositions and the operationalized definition only assesses proposition 2. Our arguments in the paper apply to both definitions of plausibility except for understandability, which uses the convincingness definition.

We will add the above clarification as a footnote in the definition of plausibility in Sec 2.

**3-3 Review Response2:**

qA5o

**3-4 Reaction To Review2:**

We thank the reviewer for thoughtful and supportive comments.

1.For the obvious nature of our position, we explain why plausibility can be a confusing concept in response #1 to Reviewer ybgB

2.We thank the reviewer for the reminder. We will add the task, key assumptions, and Conjecture 1 (currently in Appendix F) in the main text for Theorems 1 & 2

3.“Conjecture 1 a strong questionable assumption”
We apologize for the confusion and appreciate the opportunity for clarification. Conjecture 1 is a formal description of Premise 1, whose references and empirical evidence are provided in Line 231. We also empirically tested the Conjecture 1 assumption in Appendix E.1

4.On plausibility “might be used …to understand an AI output”, we agree and this doesn’t contradict with our position. To clarify using the analogy in our response #1 to Reviewer ybgB:
Our position isn’t meant to discourage AI models from learning plausible features (analogous to patients improving health and receive positive report). We are against the behaviour of encouraging XAI to present as plausible features as possible (analogous to incentivizing doctors to generate as many positive reports as possible)

5. “Paper doesn’t consider advanced uses of AI” As long as the explanation for advanced use of AI falls within the scope and definition of plausibility, our work can apply to the mentioned scenarios. We will add a sentence for it in Conclusion

6. “focuses on one-time binary decision problems”. We acknowledged it as a limitation on pg36 Line1168

7.“down to different definitions of plausible”, Yes, we provide a discussion in Line1248-53. “eqtn (1) the only way to define plausibility”, please refer to our response #2 to Reviewer YLNi

8. “without also judging probability of correctness” Because the ground truth is unknown in real-world tasks. The clues to judge the probability of prediction correctness is from AI’s instance-specific information, such as explanation or uncertainty estimate

**3-5 Review Response3:**

ybgB

**3-6 Reaction To Review3:**

We thank the reviewer for thoughtful summary and constructive comments.

1.On Weakness, indeed, Sec 3.1 or the motivating counterexamples are enough to refute the alternative. Still, the confusion of plausibility criterion persists in XAI, evidenced by statistics in Line 80-4. We also showed roots of the confusion in Sec 3.1: Although people love to see plausible explanations from XAI, just like patients love to receive positive reports from doctors, using plausibility as the XAI criterion is like incentivizing doctors to produce more positive reports. This intuitive reason (doctors should not be incentivized to give more positive reports) can be confused with the patient’s motive to pursue health (thus more positive reports) in the XAI context. Our paper provides a comprehensive examination to help more people clarify their confusion. Our examination also identifies new proper ways in XAI design and evaluation, stated in the three findings in Sec 1.

2.On format issue, we thank the reviewer for pointing this out. We resorted to using \vspace to remove the excessive empty spaces between subsections and compress spaces from previous pages to avoid excessive blank space (latex automatic adjustment tends to start a new page and won’t break a paragraph from a new subsection in half to fill up the previous page). We sincerely apologize for our mistake in formatting and for our neglect to communicate this issue of excessive space to the conference committee. We hope our paper won’t be rejected due to this formatting issue. As Reviewer YLNi says, “Evaluation criteria guide the future direction of the XAI field”. If XAI community is using a wrong criterion and loses the timely opportunity to critically discuss it, it will impede XAI scientific development, and the negative social impacts of using the wrong criterion may persist.

We confirm that we will remove all vspace in the camera-ready, if accepted, and ensure full compliance with the NeurIPS formatting guidelines.

---

### Meta-Review · Area_Chair_9xij · 2025-09-12

**Rating:** 6
**Confidence:** 4

**Strengths:**

The paper addresses a important and timely topic for the XAI community, arguing forcefully against the use of plausibility as a primary evaluation criterion. Reviewers YLNi and qA5o found the arguments to be well-supported by a comprehensive analysis that includes theoretical proofs, experiments in the appendices, and references to prior literature. The paper's presentation was praised for being clear and accessible, with Figure 1 highlighted as an effective encapsulation of the core argument. The work thoroughly considers and refutes alternative positions, proposing constructive alternatives for how plausibility could be used as an intermediate tool rather than an end goal.

**Weaknesses:**

A weakness noted by reviewers is that the paper's core argument can be perceived as intuitive or obvious, potentially limiting its novelty. Reviewer qA5o questioned if the extensive theoretical and experimental appendices belong more in a technical track than a position paper. Additionally, the paper was criticized for not sufficiently considering advanced, interactive uses of AI, instead focusing on one-time binary decisions. Some arguments lacked citations for specific alternative views, and the formal definition of plausibility was questioned regarding whether it fully captures what makes an explanation convincing to humans. A significant administrative weakness was raised by Reviewer ybgB, who identified formatting issues due to a modified LaTeX template, considering it grounds for desk rejection.

**Questions:**

Several reviewers acknowledged the importance of your topic but questioned the novelty of the core premise, with one noting that "a plausible explanation is not necessarily a correct one" may seem obvious.
Could you therefore clarify: Is your primary contribution the identification of this problem, or is it the rigorous demonstration of its consequences and the provision of a formal framework to steer the field away from it?

**Thoroughness:**

3

---

### Decision · Program_Chairs · 2025-09-26

Reject